# Grid cells are modulated by local head direction

Klara Gerlei [1], Jessica Passlack[1,5], Ian Hawes [1,5], Brianna Vandrey[1], Holly Stevens[1], Ioannis Papastathopoulos [2,3] & Matthew F. Nolan [1,4 ✉]

Grid and head direction codes represent cognitive spaces for navigation and memory. Pure grid cells generate grid codes that have been assumed to be independent of head direction, whereas conjunctive cells generate grid representations that are tuned to a single head direction. Here, we demonstrate that pure grid cells also encode head direction, but through distinct mechanisms. We show that individual firing fields of pure grid cells are tuned to multiple head directions, with the preferred sets of directions differing between fields. This local directional modulation is not predicted by previous continuous attractor or oscillatory interference models of grid firing but is accounted for by models in which pure grid cells integrate inputs from co-aligned conjunctive cells with firing rates that differ between their fields. We suggest that local directional signals from grid cells may contribute to downstream computations by decorrelating different points of view from the same location.

[1] Centre for Discovery Brain Sciences, University of Edinburgh, Edinburgh EH8 9XD, UK. [2] School of Mathematics, Maxwell Institute and Centre for Statistics, University of Edinburgh, Edinburgh EH9 3FD, UK. [3] The Alan Turing Institute, 96 Euston Road, London NW1 2DB, UK. [4] Simons Initiative for the Developing Brain, University of Edinburgh, Edinburgh EH8 9XD, UK. [5]These authors contributed equally: Jessica Passlack, Ian Hawes. ✉email: mattnolan@ed.ac.uk

Spatial cognition and memory rely on neural representations of location and head direction[1–4]. These representations include grid codes, in which neurons are active at the vertices of a grid of tessellating triangles that cover the environment an animal explores[5], and head direction codes, in which neuronal activity is tuned to head direction[6]. Effective navigation and memory require that these codes are integrated, but whether neurons that use grid codes to represent location also provide local information about head direction is unclear.

Pure grid cells and conjunctive cells found in medial entorhinal cortex (MEC) both generate grid codes. The activity of pure grid cells is thought to depend only on the position of the animal and to lack selectivity for head direction or other navigational variables[7,8]. We refer to this as omnidirectional firing (Fig. 1a). In contrast, conjunctive cells have grid firing fields that manifest only when an animal moves in a particular direction[8]. We refer to this as unidirectional firing (Fig. 1a). This distinction is maintained in established models for grid firing, which predict either omnidirectional firing, as suggested for pure grid cells, or selectivity for a single direction, as described for conjunctive cells[9–15]. Analyses of the coding properties of pure grid cells make similar assumptions[16,17].

However, a neuron's firing could instead be tuned to multiple head directions, in which case the frequency of its firing as a function of head direction would have multiple modes. We refer to this as multidirectional firing (Fig. 1a). The possibility that pure grid cell activity is modulated in this way by head direction has received little attention.

Here we investigate the modulation of pure grid cell firing by head direction globally, or in other words across all firing fields, and locally, at the scale of individual fields. We show that, when considered globally, pure grid cells are usually active in all head directions but nevertheless show selectivity for head direction. We find that this multidirectional selectivity results from local modulation by head direction, with individual firing fields having one or more different preferred directions. This local directional modulation is not predicted by previously proposed continuous attractor or oscillatory interference models for grid firing but can be explained by models in which pure grid cells integrate inputs from conjunctive cells with fields that differ from one another in their maximum firing rates. This rich multidirectional modulation suggests that, in addition to generating a metric for space, pure grid cells could also provide downstream neurons with local viewpoint information.

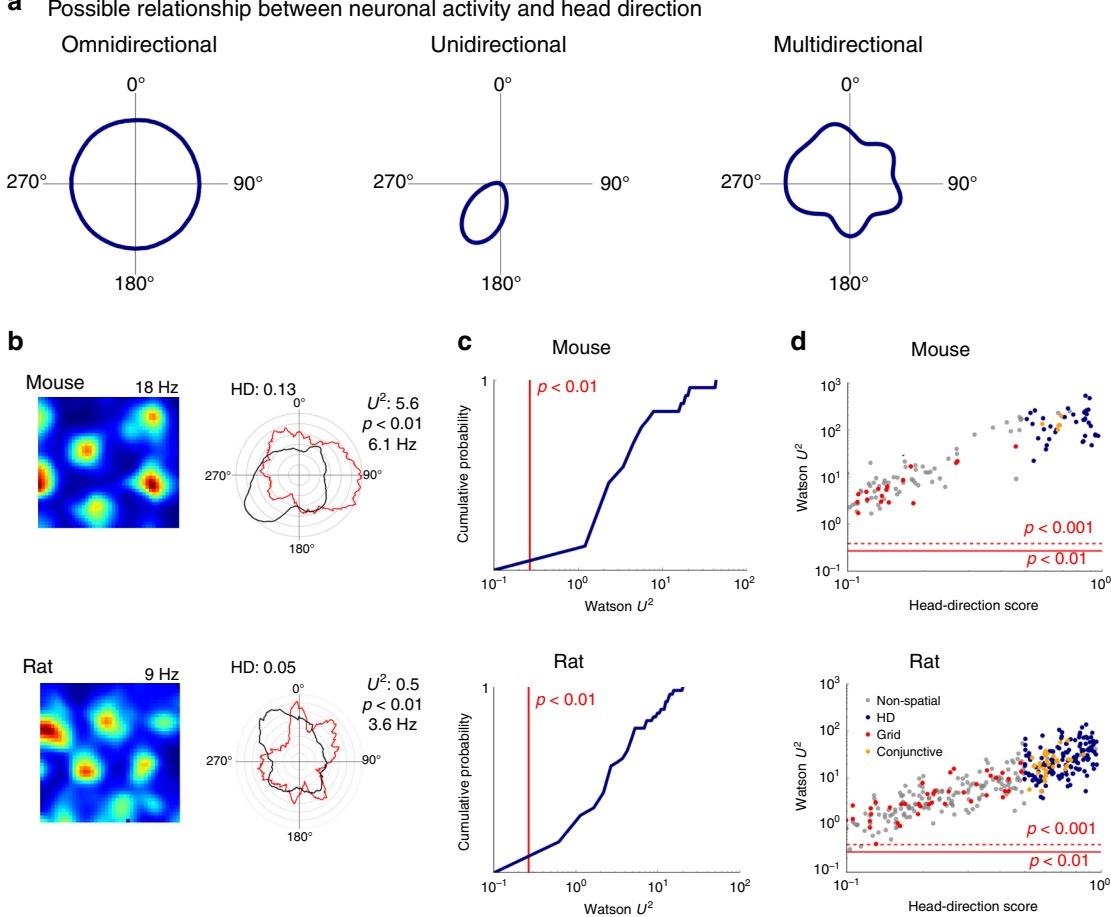

**Fig. 1 Grid cell firing is modulated by head direction. a** Schematics of omnidirectional, unidirectional and multidirectional tuning. **b** Examples of firing rate maps (left) and head direction histograms (right) for grid cells from mice (upper) and rats (lower). Polar histograms show binned counts of head direction across all video frames (black) and the mean firing rate for each head direction bin (red). The head direction score (HD), two-sided Watson $U^2$ test statistic and corresponding significance estimate and maximum firing rates for each cell are indicated adjacent to the polar plot. **c** Cumulative probability of the two-sample Watson $U^2$ test statistic comparing the distribution of head directions when each cell fired with the distribution of head directions for the entire trajectory within the recording session (mice: $p < 0.001$ for 34/34 cells, rats: $p < 0.01$ for 68/68 cells). **d** Two-sample Watson test statistics plotted as a function of head direction score for pure grid cells (red), head direction cells (blue), conjunctive cells (orange) and other cells (grey). The red lines indicate significance levels $p < 0.001$ (dashed) and $p < 0.01$ (solid).

## Results

**Grid cell firing is modulated by head direction**. To investigate directionality of grid cell firing, we analysed the activity of neurons recorded with tetrodes targeted to the MEC of 8 mice exploring an open field arena for $25.1 \pm 3.8$ min/session. We recorded from 300 neurons across 99 recording sessions (mean ± SD: $12.38 \pm 2.97$ sessions; range: 8–16 sessions per animal). Tetrode locations in the MEC were confirmed for seven of the recorded mice. We identified 13% of the recorded neurons (39/300) as having grid-like spatial firing fields using a metric based on the rotational symmetry of the autocorrelogram of their firing rate map[5] (see 'Methods'). Evaluation of the directional firing of these neurons, using a metric that detects unimodal directional bias (cf. Fig. 1a; see head direction score in 'Methods'), distinguished conjunctive cells, which were tuned to a single head direction ($n = 5$) from pure grid cells ($n = 34$), which across the whole environment were active during head orientation in all directions (Supplementary Note 2).

Because a low directional bias might nevertheless be compatible with a neuron encoding multiple directions (cf. Fig. 1a), we also assessed directional tuning by comparing the distribution of head directions when action potentials fired, with the distribution of directions throughout the animal's trajectory. A commonly used way to visualize these distributions is using 'classic' head direction plots (Fig. 1b). The differences between the distributions were significant for all pure grid cells ($p < 0.001$, $n = 34$, two-sample Watson $U^2$ test; Fig. 1c, d), providing strong evidence against the null hypothesis of independence between pure grid cell firing and head direction (cf. Fig. 1a). To validate this result and to test whether this apparent multidirectional tuning extends across species, we analysed a previously published rat data set ($n = 68$ pure grid cells)[8]. Although the duration of recordings in the rat experiments was shorter than for the mouse data ($12 \pm 4.9$ min, $t = 11.68$ $p = 2.18 \times 10^{-20}$, $t$ test), which may reduce the power of tests for directionality, analysis of these data indicated that, rather than being omnidirectional, firing of rat pure grid cells was also tuned to multiple directions (Fig. 1b–d).

The results of the Watson $U^2$ test indicate a directional preference in the firing of pure grid cells, but this could arise from variation in the running direction in different parts of a firing field, rather than from true dependence of firing on head direction (Fig. 2a). This scenario for emergence of apparent directionality from solely location-specific firing, previously referred to as the 'distributive hypothesis', accounts for apparent directional tuning of place cells in certain environments (cf. ref. [18]). To address whether the distributive hypothesis accounts for the directional firing of pure grid cells, we compared spike rate binned by the animal's head direction for the observed data with shuffled data sets generated by allocating spikes to the behavioural trajectory according to the location-dependent average firing rate (Fig. 2b and Supplementary Figs. 1 and 2). We refer to the resulting plots as 'distributive plots' as they enable assessment of a neuron's directional tuning in comparison to predictions made under the distributive hypothesis. When we tested whether the experimental firing rate for each directional bin in the distributive plots differed significantly from the shuffled data (threshold $p < 0.05$ after correcting for multiple comparisons made across bins), we found $7.3 \pm 3.9$ and $4.1 \pm 4.5$ significant bins out of 20 bins/cell for mice and rats, respectively (Fig. 2c). For all pure grid cells from mice ($n = 34/34$ cells) and most pure grid cells from rats ($n = 56/68$ cells), there were more significant bins than expected based on the shuffled data (for both data sets $p < 10^{-16}$, $U > 10$, Mann–Whitney $U$ test vs the shuffled data; Fig. 2c). We obtained similar results when we analysed firing as a function of movement direction rather than head direction, although the

effects of movement direction were smaller (Supplementary Fig. 3a). In contrast to the unimodal directional tuning of conjunctive cells[8], the directionally binned firing of pure grid cells had multiple peaks and troughs. The orientation of the peaks differed substantially between pure grid cells indicating that they were not driven by common external cues (Supplementary Fig. 4). Variation in running speed between different parts of the environment is also unlikely to account for directional tuning as, in agreement with previous studies[19], firing of most pure grid cells had speed scores below the threshold previously used to identify speed cells (cf. refs. [19–21]; median speed score for mouse grid cells = $0.068 \pm 0.18$, $n = 34$, $n = 26/34$ with speed score <0.1; median for rat grid cells from the rat data set = $0.038 \pm 0.048$, $n = 68$, $n = 60/68$ with speed score <0.1; see Supplementary Fig. 5a and Supplementary Note 2) and directional tuning was independent of a neuron's speed score (Supplementary Fig. 5b).

Together, these analyses indicate that firing of pure grid cells has a multimodal directional structure that is qualitatively distinct from the unidirectional tuning of conjunctive cells.

**Pure grid fields are locally modulated by head direction**. If firing by pure grid cells encodes head direction, then we expect this to also manifest at the level of individual firing fields. To test this, we isolated spikes from each field using a watershed algorithm (44 fields isolated from 13 pure grid cells in 4 mice and 83 fields from 25 pure grid cells in 5 rats; Fig. 3a) and analysed directional firing separately for each field (Fig. 3b, c). We used the watershed algorithm to avoid potential bias from manual selection of fields and only selected cells for further analysis when the algorithm identified at least two fields.

Individual fields demonstrated clearer directional peaks than for the arena as a whole suggesting a greater degree of directional modulation at smaller spatial scales (cf. Figs. 1b and 3c). To assess whether these peaks could have arisen by chance, we generated distributive plots for each field (Fig. 3b, c, Supplementary Fig. 6). For almost all pure grid cells, the firing rate in at least one directional bin of the plot differed significantly from the corresponding shuffled data after correcting for multiple comparisons across bins (mice: 12/13 cells; rats 24/25 cells). The number of significant bins per field was substantially greater than predicted by the shuffled data (mice: $4.3 \pm 3.2$ bins/field; rats: $2.1 \pm 2.8$ bins/field, for mice and rats $p < 10^{-16}$, $U > 8$, vs shuffled data, Mann–Whitney $U$ test; Fig. 3d) and did not correlate with bias in the behavioural head direction within the field (Supplementary Fig. 7). The proportion of fields per grid cell with directional bins that remained significant after correcting for multiple comparisons was $84.6 \pm 27.1\%$ for mice (38/44 fields) and $61.7 \pm 28.2\%$ for rats (47/83 fields). This head direction dependence at the level of individual fields could not be explained by speed dependence of neuronal firing (Supplementary Fig. 5c). In contrast to unimodal tuning of conjunctive cells and head direction cells, when fields from pure grid cells had significantly modulated bins within their distributive plot, their relative orientation was often consistent with multidirectional tuning (cf. Fig. 3c and Supplementary Note 2). Although the presence of multiple preferred directions in the fields of pure grid cells limits the ability to directly compare the strength of directional modulation with unidirectional conjunctive cells, each cell type nevertheless demonstrated a similar proportion of directionally modulated fields (Supplementary Fig. 8). Together, these analyses demonstrate directional modulation of pure grid cell firing fields that is not explained by the distributive hypothesis (cf. Fig. 2a) or by potential confounds introduced by speed-dependent neuronal firing.

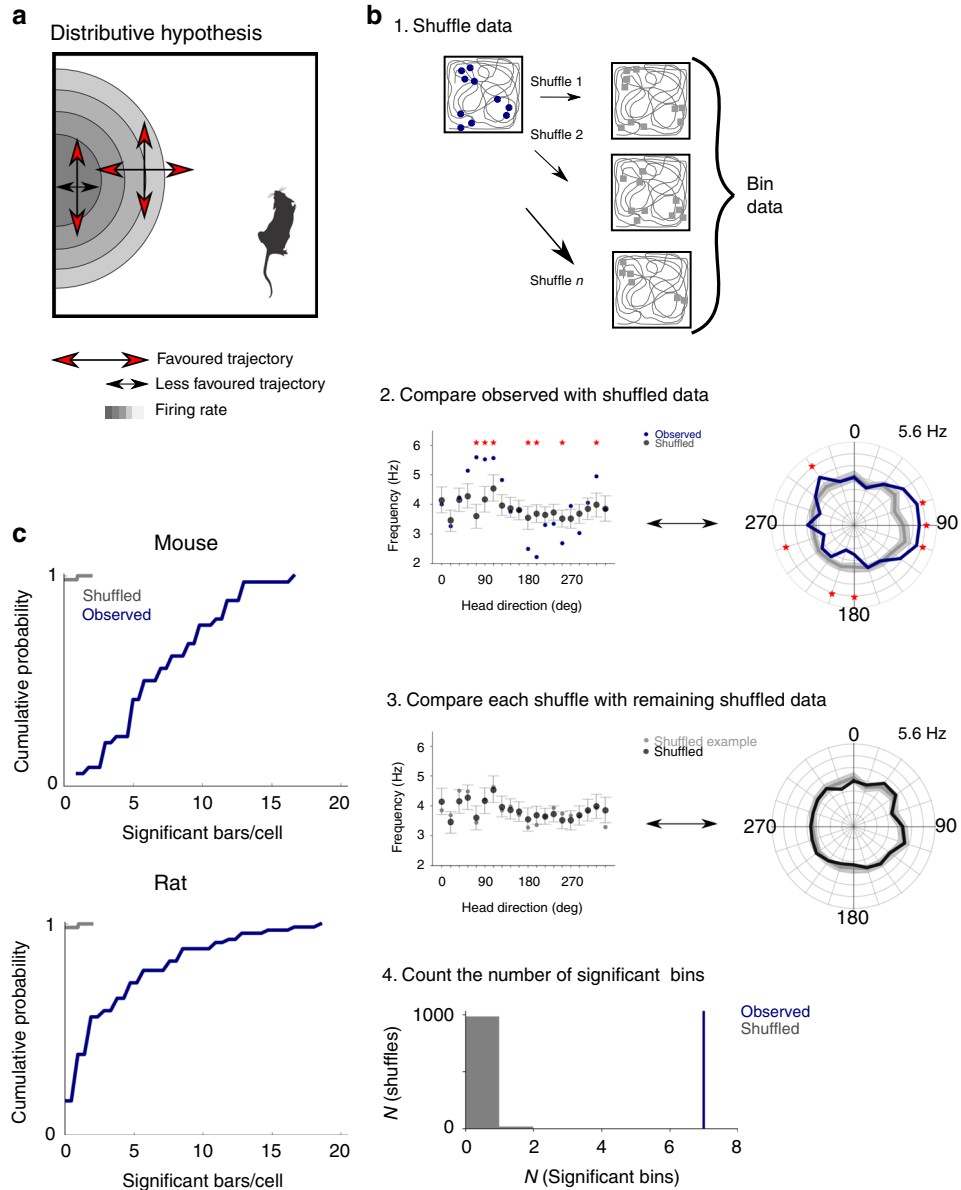

**Fig. 2 The distributive hypothesis does not account for directional modulation. a** Schematic of directional tuning caused by behavioural bias that can be explained by the distributive hypothesis[18]. Red arrows indicate head directions likely to be more sampled relative to the directions indicated by the black arrows. Differences in sampling directions at the centre and edge of the field can lead to an apparent directional bias (Mouse schematic from scidraw.io (https://doi.org/10.5281/zenodo.3910057, https://creativecommons.org/licenses/by/4.0/). **b** (1) Shuffled spike locations were generated by allocating spikes to the trajectory according to probabilities determined by the spatial firing rate map. Circles represent recorded spikes and squares represent shuffled spikes. (2) Observed and shuffled spike rates binned by head direction (bin width 18°, distributive plot) and visualized either with Cartesian (left) or polar coordinates (right). Error bars/shaded areas indicate the 90% confidence intervals of the shuffled data and the measure of centre is the mean. Asterisks indicate bins in which the observed data differs significantly from the shuffled data ($p < 0.05$, two-tailed $p$ value calculated from the shuffled distribution and corrected for multiple comparisons with the Benjamini–Hochberg procedure). (3) As for step 2 but comparing individual shuffles to the overall shuffled distribution. (4) The number of significant bins in the observed data (determined as in step 2) and the shuffled data (determined by repeating step 3 for all shuffles). The example data are from the cell in Fig. 1b. **c** The distributions of the number of significant bins per cell differed significantly between observed and shuffled data ($n = 34$ grid cells from mice, $p = 6.5 \times 10^{-24}$, $U = 10.08$; $n = 68$ grid cells from rats, $p = 2.4 \times 10^{-29}$, $U = 11.25$, two-sided Mann–Whitney $U$ test).

Inspection of individual fields from the same cell suggests that their directional modulation differs from one another (Fig. 3c). If this is the case, then directional tuning of fields from the same cell would on average show little or no correlation. In contrast, if directional modulation is non-local, then head direction tuning of fields from the same cell should be correlated. We found that fields from the same pure grid cell on average showed weak or no correlation in their directional firing (median correlation for

mice: $0.016 \pm 0.36$; for rats: $0.12 \pm 0.28$; Fig. 4a, b), consistent with local directional modulation. In contrast, correlations between fields from similarly sampled conjunctive cells were clearly detectable (median correlation for mice: $0.82 \pm 0.067$; rats: $0.72 \pm 0.26$; Fig. 4a, b). Furthermore, when comparing fields from the first and second half of the recording session (Fig. 4c), correlations were detectable for the same field but were absent between different fields from the same cell (median correlation

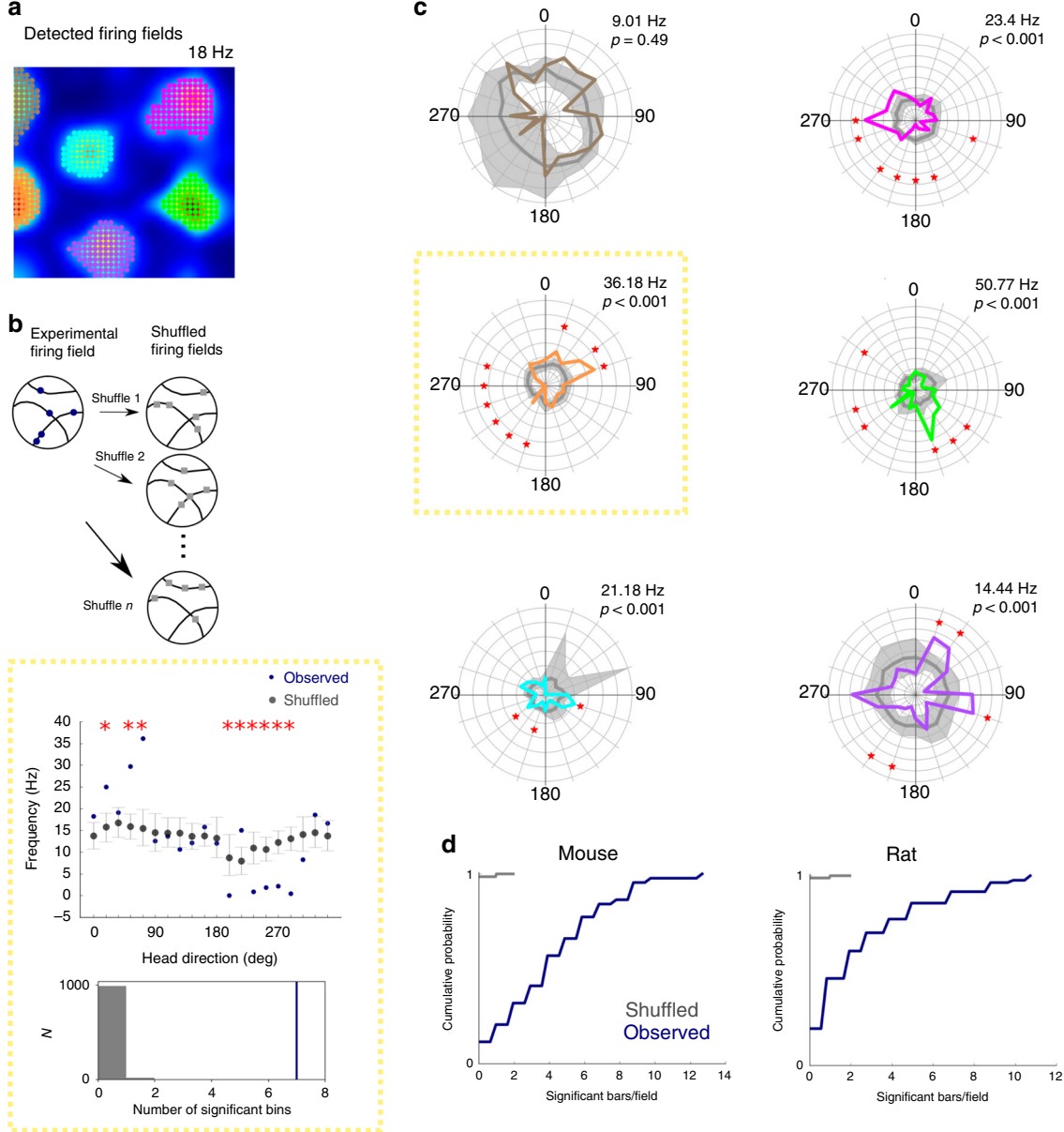

**Fig. 3 Individual firing fields are modulated by head direction. a** Firing rate map of the mouse grid cell from Fig. 1b with colour-coded automatically detected firing fields. **b** Schematic of shuffling method (upper), example directional firing rate histogram for observed and shuffled spikes (middle) for the highlighted field from **c** (yellow box) and distribution of the number of significant bins from the shuffled data (grey) and the observed data (blue) (lower). The error bars represent the 90% confidence interval of the shuffled distribution and the measure of centre is the mean. Asterisks indicate bins in which the observed data differs significantly from the shuffled data ($p < 0.05$, two-tailed $p$ value calculated from the shuffled distribution and corrected for multiple comparisons with the Benjamini–Hochberg procedure). **c** Distributive plots for each firing field (coloured according to **a**, field in yellow box also shown in **b**). The maximum firing rates are shown above the head direction plots. Significantly directional bins ($p < 0.05$, two-tailed $p$ value calculated from the shuffled distribution and corrected for multiple comparisons with the Benjamini–Hochberg procedure) are marked with an asterisk (*). **d** The number of directional bins (out of 20 bins) that differ significantly (threshold $p < 0.05$ after correction for multiple comparisons) from the shuffled data. The numbers of significant bins differed between observed and shuffled data (mice: $n = 44$ fields from 13 grid cells, $p = 5.1 \times 10^{-23}$, $U = 9.88$, rats: $n = 83$ fields from 25 grid cells, $p = 1.2 \times 10^{-18}$, $U = 8.81$, two-sided Mann–Whitney $U$ test).

for mice: $-0.025 \pm 0.31$; for rats: $-0.087 \pm 0.39$), and the correlations between different fields differed from the within-field correlations (for mice: $D = 0.38$, $p = 3.3 \times 10^{-6}$; for rats: $D = 0.53$, $p = 0.018$; Kolmogorov–Smirnov test). The relatively small difference here for data from rats likely reflects the smaller number of spikes available for the analysis. Directional firing remained uncorrelated between fields when considering only fields adjacent to the walls of the arena, or only fields in the centre of the arena, indicating that the location dependence of

directional tuning is also not related to the proximity of fields to the borders of the arena (Fig. 4d and Supplementary Fig. 9). Examination of correlations between fields as a function of their distance from one another also did not reveal any local patterns in their directional modulation (Supplementary Fig. 10).

Together, these analyses indicate that individual firing fields from pure grid cells are modulated by head direction, with many fields having multiple preferred directions. As directional modulation within each grid field is independent of direction modulation

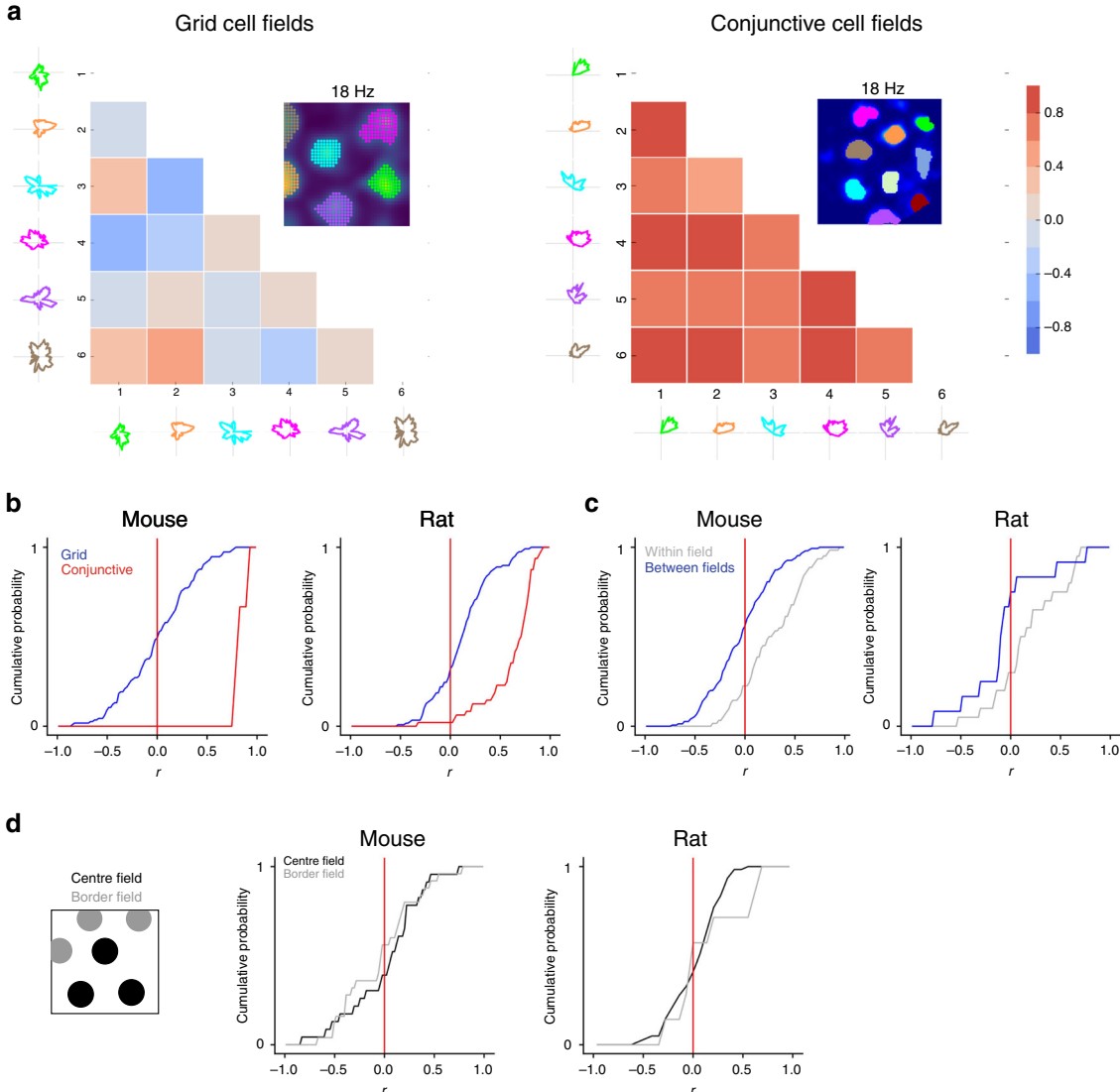

**Fig. 4 Directional modulation depends on location. a** Example of pairwise correlation of directionally binned firing of a pure grid cell (left) and a conjunctive cell (right). **b** Distribution of the coefficients obtained by Pearson correlation between different fields from the same grid or conjunctive cells calculated as in **a**. The distribution for grid cells differed from the distribution for conjunctive cells (for mice: $D = 0.98$, $p = 1.5 \times 10^{-8}$; for rats: $D = 0.72$, $p = 2.2 \times 10^{-16}$; two-sided Kolmogorov–Smirnov test). **c** Correlations between firing rate histograms for the same field (blue) and between different fields (grey) generated from data from the first and second halves of each session. The distribution of correlation coefficients for the within-field correlations differed from the between-field correlations (for mice: $p = 3.3 \times 10^{-6}$, $D = 0.38$; for rats: $p = 0.018$, $D = 0.53$; two-sided Kolmogorov–Smirnov test). **d** Border and centre fields were defined based on whether they contacted the border of the enclosure (left). Cumulative histograms show the distribution of correlation coefficients calculated as in **b** but with central and border fields shown separately.

of other fields from the same grid cell, these data indicate that directional modulation of grid cell firing depends on location.

**Stability of directional modulation.** To assess the stability of directional modulation of pure grid cells, we first quantified the correlation between global classic head direction plots generated from the first and second half of each recording session (Fig. 5a–c). Across the population of pure grid cells from mice, the correlation coefficients were skewed to positive values (median Pearson correlation = $0.47 \pm 0.39$) and differed significantly from zero (Fig. 5c). To address the possible contribution of biases in the running direction between different parts of the firing field (cf. ref. [18]), we compared correlations for shuffled and observed data from distributive plots for the two halves of each recording session (Fig. 5d). We found that 16/34 mouse grid cells had percentile scores in the top 95% of correlation scores generated from the shuffled data. For

individual fields, their classic head direction plots were also positively correlated (median Pearson correlation = $0.29 \pm 0.31$; Fig. 5e–g) and 8/44 fields had correlation scores in the 95th percentile of the shuffled data (Fig. 5h). These results are similar to predictions from 'ground truth' data generated by simulations of spiking models of directionally modulated pure grid cells (see below and Supplementary Fig. 14). These simulations predicted correlation scores between the first and second half of each session for global (median = $0.38 \pm 0.27$) and local (median = $0.58 \pm 0.30$) directional modulation that were comparable to the experimental data (Supplementary Fig. 14). They also predicted a similar proportion of correlation scores in the 95th percentile of the corresponding scores generated from the shuffled data (all cells: 12/19; fields: 29/106; Supplementary Fig. 14). Thus global directional modulation of pure grid cell firing appears stable at the time scale of single recording sessions.

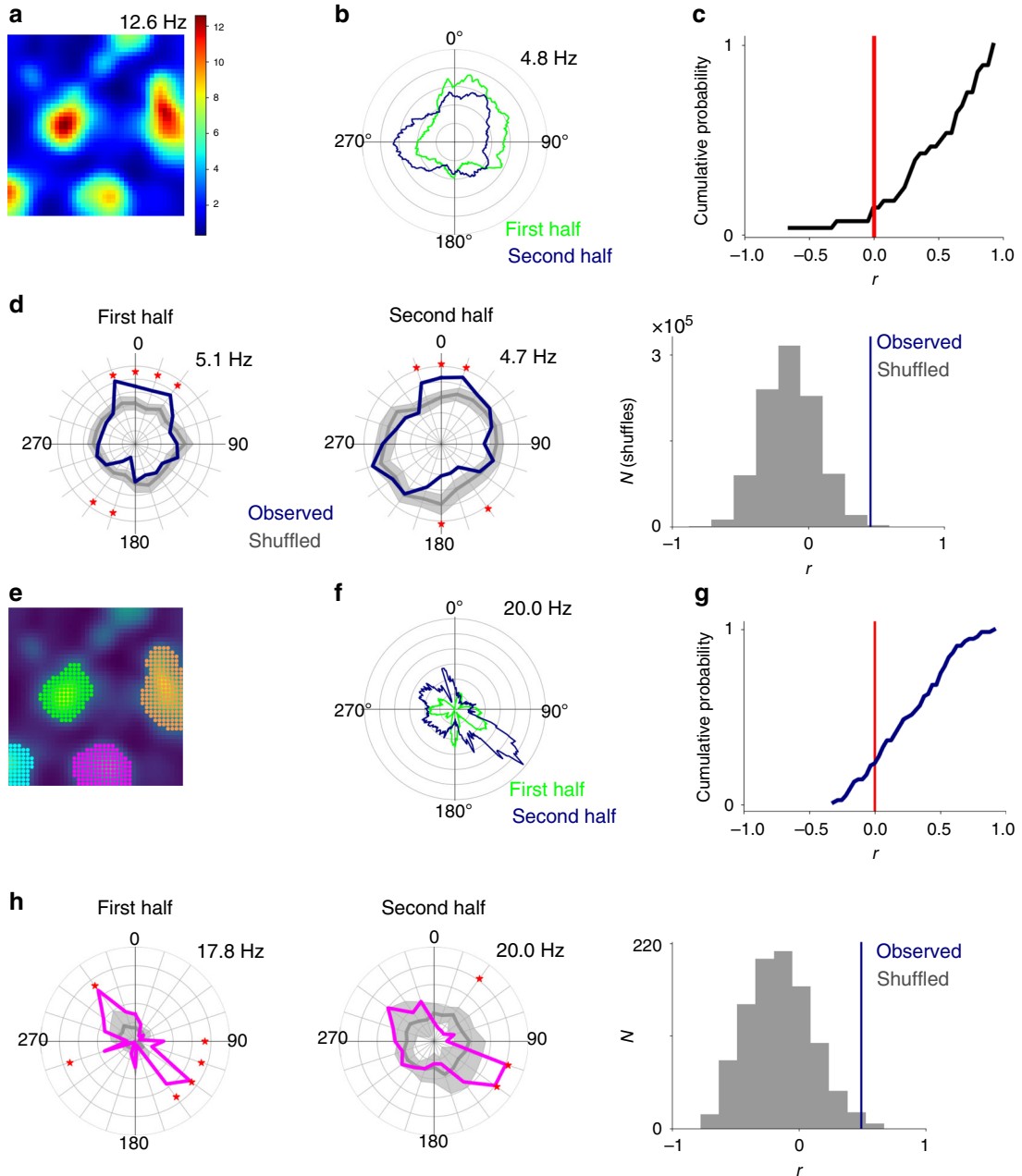

**Fig. 5 Stability of head direction preferences. a**, **b** Example firing rate map (**a**) and classic head direction plots for the first (green) and second half (blue) of the recording session (**b**). **c** Cumulative probability of Pearson correlation coefficients (mean = 0.47 ± 0.39) calculated as in **b** for all pure grid cells from mice. The median of the distribution of correlation coefficients differed from 0 (Wilcoxon one-sample signed-rank test, $p = 4 \times 10^{-5}$, $T = 23$). **d** Distributive head direction plots for spikes from the first and second half of the session for the neuron in **a** and comparison of the correlation coefficient for the observed data (blue) with the distribution of coefficients generated from the shuffled data (grey). **e**, **f** Firing rate map colour coded by field (**e**) and corresponding classic head direction plot for the first (green) and second half (blue) of the recording session (**f**) for the pink grid field shown from **e**. **g** Distributions of Pearson correlation coefficients for all fields calculated as in **f**. The median (0.29 ± 0.31 for mice) differed from zero (two-sided Wilcoxon test $p = 3.37 \times 10^{-9}$, $T = 321$, $n = 44$). **h** Distributive head direction plots for the first and second half of the session for the field in **f** and comparison of the correlation coefficient for the observed data (blue) with the distribution of coefficients generated from the shuffled data (grey).

**Models for directional modulation of grid fields**. Our analyses imply that spatially modulated multi-directional firing is a core feature of the activity of pure grid cells that does not appear to be predicted by existing models for grid firing[9–13,22] (Supplementary Table 1). For example, a requirement of many continuous attractor models is that each grid cell has a single preferred direction of input[9,11]. Consistent with this intuition, simulations using experimentally recorded trajectories as input to each of two previously described continuous attractor models[9,13] and two

previous oscillatory interference models[12,23] did not generate directionally modulated grid firing fields (Fig. 6 and Supplementary Fig. 11). Because these simulations used real trajectories, their results also provide further evidence against the hypothesis that directional tuning identified in our analyses of experimental data is a result of movement-related variables.

What mechanisms might then explain the local directional firing of pure grid cells? We reasoned that three ingredients may be important (Fig. 7a). First, upstream directional grid signals,

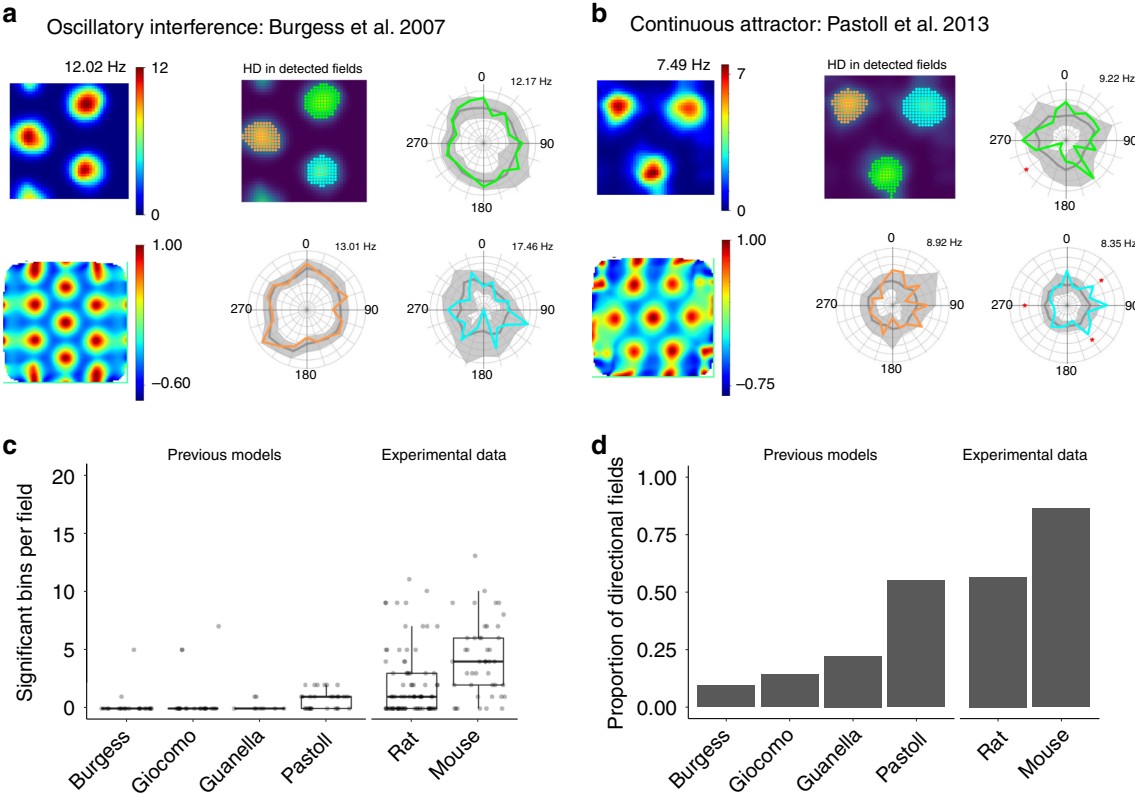

**Fig. 6 Existing models do not account for directional modulation of grid firing. a, b** Firing rate map (upper left), autocorrelogram (lower left), detected fields (upper centre) and distributive plots for the colour-coded fields for two types of grid models ran on our trajectory data. **c** Number of significant bins per field compared across previous grid cell models and experimental data. Each of the previous models (Burgess, Giocomo, Guanella and Pastoll[9,12,13,23]) differed significantly from the mouse data ($p = 5.7 \times 10^{-8}$, $p = 9.6 \times 10^{-5}$, $p = 3.4 \times 10^{-8}$, $p = 4.2 \times 10^{-7}$, respectively, one-way ANOVA with Games–Howell test), and each of the models differed significantly from the rat data, with the exception of Giocomo ($p = 6.7 \times 10^{-4}$, $p = 0.32$, $p = 4.1 \times 10^{-5}$, $p = 6.1 \times 10^{-3}$, respectively). **d** Proportion of directionally modulated fields compared across existing grid cell models and experimental data.

which may originate from conjunctive cells[8]. Second, differences between fields of the upstream cells in their mean firing rate. This is consistent with grid fields encoding positional information in their local firing rate[24]. Third, convergence of multiple upstream signals that have co-aligned spatial firing fields and different head direction preferences. This is consistent with the prominent projections from deep layers of MEC, where conjunctive cells are most frequently found, to superficial layers where pure grid cells are most frequently found[8,25–27]. Models have been proposed previously that combine the first and third ingredients to account for firing of pure grid cells (cf. refs. [15,25]). We refer to these as spatially uniform conjunctive cell input models as each of the upstream conjunctive cells' fields has similar mean firing rates. We consider here the possibility that introducing the second ingredient would account for the experimentally observed local directional tuning of pure grid cells. We refer to this new model as a spatially non-uniform conjunctive cell input model (Fig. 7a). We simulated both classes of model using an anatomically reconstructed stellate cell configured to account for experimentally recorded excitable properties and with synaptic inputs from conjunctive cells distributed across its dendritic tree (Fig. 7a and Supplementary Fig. 12). Because the strength of in vivo synaptic inputs to stellate cells is unknown, we simulated versions of the uniform and non-uniform input model with relatively low and relatively high synaptic conductances.

We found that the spatially non-uniform conjunctive cell input model accounted well for the global and local directional modulation of pure grid cell firing fields. When we simulated this architecture with experimentally recorded trajectories as

inputs, the firing of the model grid cell across an environment was directionally modulated (Fig. 7b) and each of its fields had a distinct directional modulation (Fig. 7c), resembling the experimentally recorded directional modulation of pure grid cells (Supplementary Fig. 6 and cf. Fig. 7c with Fig. 3c). The proportion of directionally modulated fields and the number of significant bins per field in their distributive plots generated by the low- and high-conductance versions of the non-uniform model spanned the range of the experimental observations from rats and mice (Fig. 7d). Similar to our experimental observations (cf. Fig. 4b), correlations between fields from the same simulated pure grid cell were weak in this model (median correlation = $-0.0061 \pm 0.11$; Fig. 8b) and between-field correlations differed from the within-field correlations (Fig. 8c), also resembling the experimental data (cf. Fig. 4c; $p = 2.2 \times 10^{-16}$, $D = 0.62$). In contrast to the non-uniform input models, versions of the models with spatially uniform conjunctive cell input generated fewer directional fields with fewer significant directional bins per fields (Supplementary Fig. 13). Despite their weaker directional modulation, the fields generated by the spatially uniform conjunctive cell input versions of the model were nevertheless positively correlated (median correlation = $0.13 \pm 0.26$), which is consistent with a lack of local directional modulation (Fig. 8b), and the between-field correlations did not differ from the corresponding within-field correlations (Fig. 8b; $p = 0.14$, $D = 0.13$). Thus integration of input from co-aligned conjunctive cells can account for the local direction selectivity of grid cell firing, but only when the conjunctive cells have fields that differ from one another in their mean firing rates.

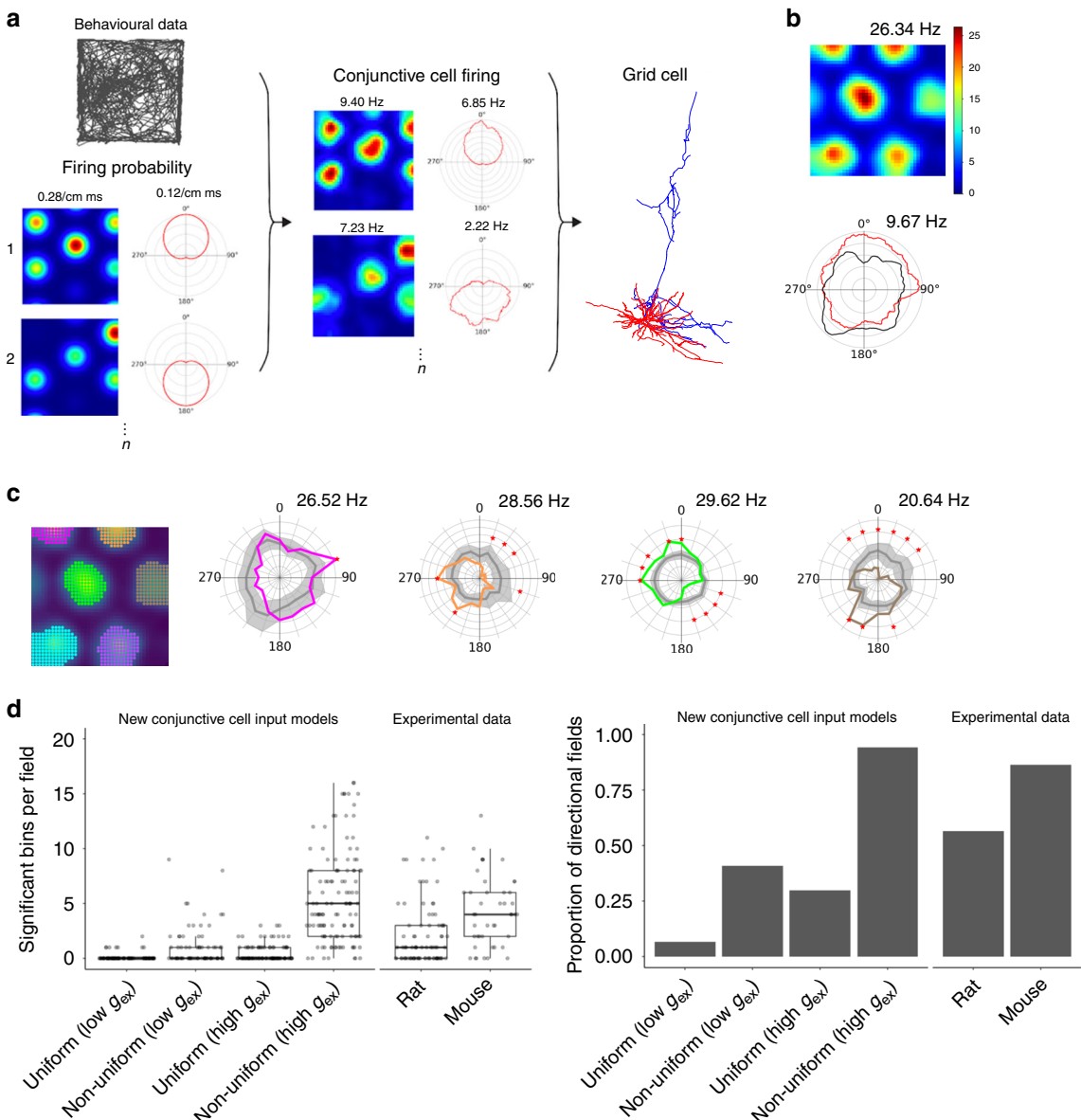

**Fig. 7 Integration of non-uniform conjunctive cell inputs accounts for directional modulation. a** Rate map and head direction polar histograms of conjunctive cell spike firing probabilities (with the peak probability listed above) were convolved with the head direction and position of experimentally tracked mice to generate conjunctive cell firing fields (peak firing rates indicated above). Firing times of the simulated conjunctive cells were used to trigger synaptic input to a compartmental model of a stellate cell. **b** Representative rate-coded firing fields and head direction histograms from a simulation with five conjunctive cell inputs. **c** Directionally binned firing rate for selected fields colour coded according to the accompanying rate map. Grey lines are the corresponding shuffled distributions. Asterisks indicate bins in which the observed data differs significantly from the shuffled data ($p < 0.05$, two-tailed $p$ value calculated from the shuffled distribution and corrected for multiple comparisons with the Benjamini–Hochberg procedure). **d** Comparison of conjunctive cell input models with experimental data, showing significant bins per field (left), and proportion of directional fields (right). The number of significant bins per field in the high conductance (high $g_{ex}$) non-uniform model was greater than for the rat data ($p = 5.7 \times 10^{-9}$, one-way ANOVA with Games–Howell test) but did not differ significantly from the mouse data ($p = 0.299$), whereas the low conductance (low $g_{ex}$) non-uniform model had fewer significant fields than the mouse data ($p = 6.1 \times 10^{-6}$) but not did not differ significantly from the rat data ($p = 0.172$). Both high- and low-conductance uniform models had fewer significant bins per field than the mouse data ($p = 7.1 \times 10^{-8}$ and $p = 7.1 \times 10^{-9}$, respectively) and rat data ($p = 1.8 \times 10^{-4}$ and $p = 1.3 \times 10^{-4}$, respectively).

Because the models with uniform and non-uniform conjunctive cell inputs generate spikes through biophysically plausible integrative mechanisms, we used their outputs to further validate the analyses of our experimental data. Systematic variation in the length of the simulated recording indicated that the typical duration of the recording sessions we used experimentally is sufficient to reliably detect directional modulation at a population and local level (Supplementary Fig. 14). Detection of correlations

between the first and second half of each recording session was less reliable, with large variability in the correlation scores obtained. This is also consistent with our experimental data (cf. Fig. 5c, g and Supplementary Fig. 14g, h).

## Discussion
Our analyses show that the firing fields of pure grid cells are locally modulated by head direction. This property of grid cells is

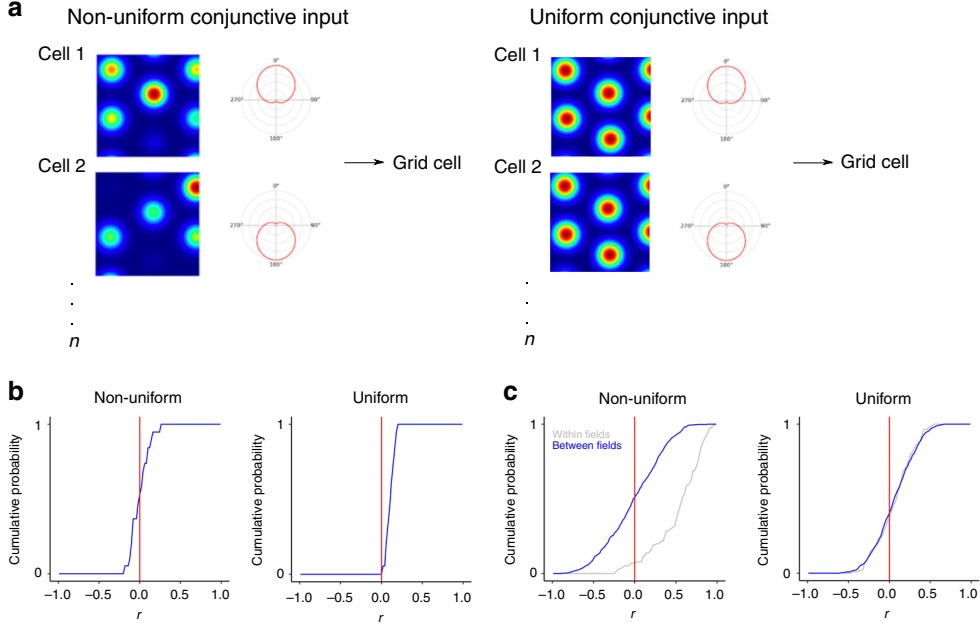

**Fig. 8 Location-dependent firing emerges when conjunctive cell activity is spatially non-uniform. a** Schematic of uniform and non-uniform conjunctive cell input models. **b** Distribution of average Pearson coefficients from correlation between fields of simulated grid cells calculated as in Fig. 4b. The distribution of correlation coefficients differed between uniform and non-uniform conjunctive input models ($p = 3.99 \times 10^{-6}$, $D = 0.21$, two-sided Kolmogorov–Smirnov test). **c** Correlations between firing rate histograms for the same field (grey) and between different fields of the same cell (blue) generated as in Fig. 4. The between- and within-field distributions differed from one another for the non-uniform model ($p = 2.2 \times 10^{-16}$, $D = 0.62$, two-sided Kolmogorov–Smirnov test) but not for the uniform model ($p = 0.14$, $D = 0.13$). The half-session between field distributions differed between the non-uniform and uniform models ($p = 7.1 \times 10^{-8}$, $D = 0.17$, KS test).

not predicted by existing models but can be accounted for by integration of signals from conjunctive cells with spatially non-uniform firing rates. Spatial signals that distinguish points of view within an environment may be critical to downstream computations, for example, pattern separation functions of the dentate gyrus and encoding of head direction by place cells.

The directional modulation of grid cell firing that we describe here is qualitatively distinct from previously described properties of conjunctive cells[8]. Whereas conjunctive cells are tuned to a single preferred head direction, we find that pure grid cells show preferred firing in multiple directions (Figs. 1–4). Whereas for conjunctive cells directional tuning of individual firing fields is strongly correlated and therefore global, for pure grid cells directional modulation differs between individual fields and is therefore local (Fig. 4). These differences suggest that directional modulation of conjunctive cells and grid cells result from distinct circuit mechanisms. They also suggest reasons why directional modulation of pure grid cells has not previously been described. For grid cells, summed directional preferences from several fields generates global multi-directional modulation (cf. Fig. 1a) that is not typically considered in analyses of directional firing and is challenging to detect with shorter duration recordings (cf. Supplementary Fig. 14).

Embedding of location-dependent directional information within grid cell firing fields has implications for the organization of spatial computations in entorhinal circuits and downstream structures. While more complex models that account for location-dependent directional modulation of pure grid cells could be envisaged (Supplementary Table 1), integration of input from conjunctive cells with spatially non-uniform firing rates provides a parsimonious explanation for the location-dependent directional modulation of grid cell firing (Fig. 7). This explanation is consistent with models in which grid codes originate in conjunctive cells[15,25,28] but differs in that in the model we propose

here each of a conjunctive cell's firing fields has a different mean firing rate. It is this difference in firing rate that leads to the location-dependent directional modulation of pure grid cell firing. According to this scenario, investigation of the mechanisms underlying the grid code would most productively focus on conjunctive rather than pure grid cells. This scheme may also resolve discrepancies between functionally and anatomically defined cell types in MEC[29]. For example, entorhinal stellate cells, which are a major input to the hippocampal dentate gyrus and are required for spatial memory[30,31], appear to be the most numerous grid cell type, but less than half of stellate cells are grid cells[32]. This functional divergence is consistent with all stellate cells implementing similar cellular computations[33] but with the emergence of grid firing patterns depending on the identity of their dominant synaptic inputs.

A key future question is how directional information encoded by pure grid cells is used by downstream neurons. Given the variability in spiking at the level of single fields, it appears unlikely that downstream neurons could read out directional signals from individual pure grid cells. However, if downstream neurons receive sufficient convergent input, then local directional modulation of pure grid cells, by disambiguating different views at the same location, may facilitate pattern separation functions of the immediately downstream dentate gyrus[34–36] or CA1 place cells[37]. For example, recently described directional selectivity of place cells is also multi-modal and location dependent[37] and could be driven by the location-dependent directional tuning of pure grid cells that we describe here. In this case, direction-dependent modulation of grid firing may be a computational feature of representations of visual and conceptual scenes[4,38,39].

## Methods

**Animals**. All animal procedures were performed under a UK Home Office project license (PC198F2A0) in accordance with The University of Edinburgh Animal

Welfare committee's guidelines. All procedures complied with the Animals (Scientific Procedures) Act, 1986 and were approved by the Named Veterinary Surgeon and local ethical review committee.

We report data from 8 mice (4 males and 4 females) from a total of 16 that were implanted with recording electrodes. For 7 of the 8 mice, the location of the tetrodes in the MEC was confirmed anatomically after experiments were complete. For the other mouse, the tetrode location could not be determined. The age of the mice when tetrodes were implanted was 7–13 weeks (mean = 10.6 ± 1.7 weeks). Animals were excluded either because the recordings did not identify grid cells ($n = 6$) or because they had to be terminated before the end of the experiment, because their implant came off ($n = 2$). The mice used were from the p038 line[40]. Before surgery, animals were group housed (3–5 mice per cage) in a standard holding room on a standard 12-h on/off light cycle (dark from 7 p.m. to 7 a.m.). After surgery, mice were single housed in a different holding room in otherwise similar conditions (average temperature 20 °C, relative humidity 50%). Mice were kept in individually ventilated cages containing sawdust, tissues, chewing sticks and cardboard tubes, which after surgery were replaced by a larger cardboard igloo. Two days after the surgery, a running wheel was placed in the cages. Standard laboratory chow and water were given ad libitum.

**Microdrive design**. We modified previous designs for 16-channel microdrives consisting of 4 tetrodes and an optic fibre[41,42]. A 21-gauge 9 mm long inner cannula (Stainless Tube & Needle Co. LTD) was attached to an EIB-16 board (Neuralynx) using epoxy (RS components 132-605). Tetrodes were made with 18 μm HML-coated 90% platinum, 10% iridium wire (Neuralynx). We connected grounding wires (1.5 cm long insulated part) to the reference and ground pins of the EIB. The four tetrodes were threaded through the inner cannula, connected to the pinholes of the EIB-16 board and fixed with gold pins (Neuralynx, EIB Pins). A 13-mm-long optic fibre stub (Plexon, PX.OPT-FS-Flat-200/230-13L) was threaded through the inner cannula between the tetrodes. The wires on the board and the optic ferrule were covered to about 2/5th of the length of the optic ferrule with epoxy and the tetrodes glued (RS components, 473-455) to 4 sides of the optic fibre. The tetrodes were cut using ceramic scissors (Fine Science Tools, Germany) to a length of 0.5 mm from the tip of the optic fibre. The tips of the tetrodes were plated in a non-cyanide gold plating solution (Neuralynx). Tetrodes were cleaned, by applying 3 1-s 4 μA pulses with the tetrodes as an anode and then plated by passing 2 μA 1-s pulses with the tetrodes as a cathode until their impedance was between 150 and 200 KΩ.

**Microdrive implantation**. General surgical procedures and stereotaxic viral injections were carried out as described previously[43]. We induced inhalation anaesthesia using 5% isoflurane/95% oxygen and sustained at 1–2% isoflurane/98–99% oxygen throughout the procedure (1 litre/min). Before implanting the drive, we injected AAV9-tre-ChR2-mCherry (Gene Therapy Center, University of Massachusetts Medical School; 800–2000 nl total injection volume, 3–5 injection sites, 200–400 nl/site) for additional opto-tagging experiments (data not shown). All animals were injected 3.4 mm lateral relative to bregma (Supplementary Table 2). For electrical grounding, we drilled two small craniotomies and implanted M1 × 4 mm screws (AccuGroup SFE-M1-4-A2) on both sides about 3.4 mm lateral and 1 mm rostral relative to bregma.

We stereotaxically lowered the tetrodes 1.5 mm into the brain, beginning 3.4 mm lateral from bregma (right hemisphere of two mice and left hemisphere of 14 mice) and targeting the dorsal third of the MEC. We sealed the outer cannula with sterile Vaseline and fixed the implant by putting dental acrylic (Simplex Rapid powder) around the drive frame and the outer cannula, wrapped the grounding wires around the grounding screws, fixed the wires with silver paint (RS components 101-5621) and applied another layer of dental acrylic to cover the skull and the grounding screws. Mice recovered on a heat mat for approximately 20 min and were given Vetergesic jelly (0.5 mg/kg of body weight buprenorphine in raspberry jelly) for 12 h after surgery.

**Open field exploration task**. All recordings from mice were performed in an open field arena consisting of a box made from a black metal frame (parts from Kanya UK, C01-1, C20-10, A33-12, B49-75, B48-75, A39-31, ALU3), with removable black metal walls, a polarizing cue (white A4 paper) on one wall, and a floor area of 1 m². A camera (Logitech B525, 1280 × 720 pixels Webcam, RS components 795-0876) was mounted on the top of the frame for motion tracking. To record head direction, we used a custom Bonsai script[44] to track red and green polystyrene balls attached to either side of the mouse's head on the left and right sides. The distance between the polystyrene balls was approximately 3.5 cm.

Mice were handled three times a week for 5–10 min for 4 weeks following surgery. For 3 consecutive days before recording, we habituated the mice for allowing them to explore the open field arena for 5–10 min. For recording sessions, mice explored the open field arena unrewarded until they covered the whole area or for a maximum of 90 min. An opto-tagging experiment was performed at the end of each recording session (data not shown). After each recording session, we lowered the tetrodes by 50 μm using the drive mechanism on the implant.

For electrophysiological recording, the 16-channel optetrode was connected to an Open Ephys acquisition board[45] and computer (HP Z440 Tower Workstation

i7, 16 GB, 512 GB SSD, Cat.: J9CO7EA#ABU) using an SPI cable (Intan Technologies, RHD2000 6-ft (1.8 m) Ultra Thin SPI interface cable C3216) and via a commutator (SPI cable adapter board, Intan Technologies C3430 and custom 3D printed holder). Signals were filtered between 2.5 and 7603.8 Hz using a second-order Butterworth filter implemented in Open Ephys. We aligned position and electrophysiology data using light pulses generated at random intervals (20–60 s) by a light-emitting diode attached to the side of the open field arena hidden from the mouse but in the field of view of the camera.

**Post recording assessment of tetrode locations**. To enable determination of tetrode locations, after the last recording day we anaesthetised mice using an isoflurane chamber and pentobarbital (100–150 μl) and applied a 2-s ~20 μA current to burn the tissue at the tip of the electrodes. We then intracardially perfused phosphate-buffered saline (PBS, Gibco, 70011044, 10 times diluted with distilled water) for 2 min, then 4% paraformaldehyde (PFA, Sigma Aldrich, 30525-89-4) in 0.1 M phosphate buffer (PB, Sigma Aldrich, P7994) for 4 min at a 10 ml/min flow rate. We left the brains in 4% PFA in 0.1 M PB for 16 h, then transferred them to 30% sucrose (Sigma Aldrich, S0389) in PBS until they sank.

We cut 50 μm sagittal sections of the fixed brains using a freezing microtome. Sections were processed to label them with primary antibody rat anti-mCherry (Invitrogen M11217, 1:1000) followed by secondary antibody goat anti-rat Alexa 555 (Invitrogen A-21434, 1:1000) and stained with either NeuroTrace 640/660 (Invitrogen N21483, 1:500) or NeuroTrace 435/455 (Invitrogen N21479, 1:500) following procedures described previously[43]. Images were taken on a Zeiss Axio Scan Z1 using a ×10 objective and visually inspected to determine the final position of the recording electrodes (see Supplementary Note 1).

**Data analyses**. Analyses were carried out using Python (version 3.5.1 in Anaconda environment 4.0) and R version: 3.3.1 (2016-06-21). All codes are available at https://github.com/MattNolanLab/grid_cell_analysis. Summary values are reported as mean or median ± standard deviation.

**Spike sorting**. To isolate spikes from electrophysiological data, we used an automated analysis and clustering pipeline based around MountainSort 3 (v 0.11.5 and dependencies)[46]. Python scripts pre-processed the data by converting Open Ephys files to mda format and organized these files together with spike sorting input parameter files. We defined the four channels of each tetrode to be in the same 'sorting neighbourhood'. We excluded broken channels identified during data acquisition.

MountainSort filtered the data from 600 to 6000 Hz using a bandpass filter and then performed spatial whitening over all 16 channels to remove correlated noise. Events with peaks three standard deviations above average and at least 0.33 ms away from other events on the same channel were detected. The first 10 principal components of the detected waveforms were calculated. A spike sorting algorithm, ISO SPLIT, was applied to the resulting feature space[46].

Cluster quality was evaluated using metrics for isolation, noise overlap and peak signal-to-noise ratio[46]. Units that had a firing rate >0.5 Hz, isolation >0.9, noise overlap <0.05 and peak signal-to-noise ratio >1 were accepted for further analysis. Any units that did not have a refractory period or hyperpolarization component of their spike waveform were discarded. These exclusions were based on visually assessing output figures generated for sorted clusters. No additional manual curation was used.

**Classification of functional cell types**. To classify recorded neurons, we used established grid and head direction scores[8]. Grid scores were defined as the difference between the minimum correlation coefficient for rate map autocorrelogram rotations of 60 and 120 degrees and the maximum correlation coefficient for autocorrelogram rotations of 30, 90 and 150 degrees[47]. The firing rate map was calculated by summing the number of spikes in each location, dividing that by the time the animal spent there and then smoothing the surface with a Gaussian centred on each location bin[34]. Autocorrelograms were calculated by shifting the binned firing rate map[34] into every possible binned position along both horizontal and vertical axes and calculating correlation scores for each of these positions. This autocorrelogram was converted into a binary array using a 20% threshold on normalized data. If the binary array had more than seven local maxima, a grid score was calculated. Subsequent parts of the analysis, where correlations between the rotated autocorrelograms were calculated, only included the ring containing six local maxima closest to the centre of the binary array, excluding the maximum at the centre. The ring was detected based on the average distance of the 6 fields near the centre of the autocorrelogram (middle border = 1.25 × average distance, outer border = 0.25 × average distance).

To calculate head direction scores, the head direction angles corresponding to the firing events of each neuron were first binned into 360 bins between 0 and $2\pi$. The obtained polar histogram was smoothed by calculating a rolling sum over a 23 degree window. For angles between −179 and 180 degrees in steps of 1 degree, d$x$ and d$y$ were calculated in a unit circle (radius = 1) as d$y$ = sin(angle) × radius$^{-1}$ and d$x$ = cos(angle) × radius$^{-1}$. To obtain the $x$ and $y$ components of the head direction vector, the head direction polar histogram was multiplied by the d$x$ and d$y$ values, respectively, and normalized to the number of observations in the polar

head direction histogram, so that $x_{\text{total}} = \Sigma(\text{d}x\ \text{HD}_{\text{histogram}}) \times \Sigma\text{HD}_{\text{histogram}}^{-1}$ and $y_{\text{total}} = \Sigma(\text{d}y\ \text{HD}_{\text{histogram}}) \times \Sigma\text{HD}_{\text{histogram}}^{-1}$. The head direction score was then calculated using the Pythagorean theorem as head direction score $= (x_{\text{total}}^2 + y_{\text{total}}^2)^{1/2}$.

We defined grid cells as cells with a grid score $\geq 0.4$, which was chosen as a conservative threshold (cf. ref. [24]). We defined head direction cells as cells with a head direction score $\geq 0.5$. We defined conjunctive grid cells as cells that passed both head direction and grid cell criteria.

**Identification and analysis of individual fields**. We identified individual firing fields using methods similar to those used previously to detect place fields[48]. The open field arena was divided into $42 \times 42$ bins, where each bin contained a smoothed firing rate value calculated by summing the number of spikes at the locations corresponding to each bin, dividing this by the time the animal spent in the bin and then smoothing the surface with a Gaussian ($e^{-x^2/2}$) centred on each bin[34]. We next identified the bin of the rate map with the highest firing rate. If the rate was higher than the average firing rate plus the standard deviation of the rest of the rate map, we then added adjacent bins if they had a firing rate $>35\%$ of the maximum firing rate within the field. We recursively added to the field further bins that satisfied these criteria with respect to the newly added bins. We accepted a detected field if it had $>45$ bins, but it was smaller than half of the arena. After successfully detecting a field, it was removed from the rate map by replacing the values with zeros, and the analysis was repeated until we found no more fields. All detected fields were visually assessed, and if a detected field appeared to be a combination of two fields or only part of a field, it was tagged as a false positive to be excluded from the analyses.

**Analysis of head direction**. Classic head direction polar plots (Figs. 1b and 5b, f) were generated by dividing the histogram of head directions when the cell fired by the histogram of head directions from the trajectory to obtain a firing rate (Hz) for each angle. Each histogram has 1 degree bins with 23 degree smoothing.

Shuffled head direction polar plots (Figs. 2, 3, 5d, h, 6, and 7) were generated by first dividing the histogram of head directions when the cell fired by the histogram of head directions from the trajectory using 20 bins of width 18 degrees to generate a directional firing rate plot. The shuffling procedure was designed to rule out the possibility that directional firing is a result of different sampling of directions in different bins of the rate map[18]. Before generating shuffled data, the cell's spatial firing rate map was used to allocate firing probabilities to each point on the trajectory such that the total probability was one and the local probability was proportionate to that of the firing rate in the given bin of the firing rate map. Shuffled spike locations were generated by random selection with replacement of locations from the trajectory of the animal using the allocated probabilities. Each shuffled data set was sorted into directional bins in the same way as the experimental data. After generating 1000 shuffled data sets, the median and 90% confidence intervals were calculated for each bin by ranking the shuffled data.

*Two-sample Watson test*: To evaluate whether head direction when the cell fired differed from the head direction of the animal during the time spent in a field, or arena, we performed Watson's two-sample test[49,50] for homogeneity on the two distributions using the R package circular (https://r-forge.r-project.org/projects/circular/). To illustrate the distributions of head directions from the trajectory and the normalized firing rate, we made smooth head directions plots (1 degree bins and 23 degree smoothing). On plots where the firing rate and trajectory histograms are shown, the trajectory histogram was re-scaled to fit the plot.

*Shuffle tests*: Analyses compared the observed data with 1000 shuffled data sets generated as described above. For each bin, the percentile position of the observed data relative to the randomized data was used to calculate a $p$ value. The 20 $p$ values within a field were corrected for multiple comparisons using the Benjamini–Hochberg procedure and the number of intervals where the corrected $p$ values were $<0.05$ were counted. To obtain null distributions, the same analyses were performed for each of the 1000 randomized data sets, where each shuffle was treated like the observed data and was compared to the distributions from the 1000 shuffles. Fields where the percentile score of the observed number of significant bins relative to the null distribution from shuffles was $>97.5$ were considered directional.

*Correlation between directional firing histograms*: To evaluate stability of firing within a session, Pearson correlation coefficients were calculated using the SciPy function scipy.stats.pearsonr for pairs of classic head direction polar plots. Comparison of pure grid cell between- and within-field correlations used fields with $>500$ spikes. To evaluate whether correlations are influenced by the distributive hypothesis, we generated shuffled head direction polar plots for each half of the session as described above for the whole session. We calculated Pearson correlation coefficients between the distributive plots for the first and second halves of the session for the experimental data and separately for all possible combinations of the shuffled data ($1000 \times 1000$ comparisons). We then determined the percentile score of the Pearson correlation coefficient corresponding to the observed data relative to the population of coefficients derived from the shuffled data.

**Rat data**. Data from rats[8] was downloaded from the Kavli Institute's online database (https://www.ntnu.edu/kavli/research/grid-cell-data). The data were

available in a format that contained the trajectory of the animal and firing times of sorted cells in MATLAB files. The MATLAB files were converted into a spatial data frame similar to the mouse data so the same analysis scripts could be used to perform all analyses. Our analyses use 32 conjunctive cells and 68 pure grid cells identified in this data set.

**Simulation of previous grid cell models**. We simulated four previous grid cell models with experimentally recorded behavioural trajectories as inputs. Code for the Guanella, Giocomo and Burgess models was adapted from ref. [51] (hosted at ModelDB, accession number: 144006), and the Pastoll model was hosted on ModelDB (accession number: 150031). We extracted spike times as outputs from the simulations and matched these to corresponding head directions and $x$, $y$ positions from the trajectory. These model results were converted into a format used by our analysis pipeline and underwent the same standardized analysis as experimental data.

**Stellate cell model**. All simulations were performed in the NEURON simulation environment[52]. Simulation code will be available at https://github.com/MattNolanLab.

The model stellate cell used a previous morphological reconstruction of a mouse MEC layer 2 stellate cell[53]. Voltage-gated sodium and potassium channels were inserted into the soma and axons (channel models from ref. [54]), hyperpolarization-activated cyclic-nucleotide-gated channels were inserted into the dendrites and soma (channel models from ref. [55]) and leak channels were inserted into all compartments. Maximum channel conductances were adjusted so that electrophysiological properties of the neuron were similar to the experimentally determined properties of stellate cells[56], with the resting membrane potential (RMP), input resistance, sag, rheobase spike peak and half-width fit within the range of experimental values[33]. The best fit for voltage threshold that could be achieved was within 30% of the mean experimentally determined values. RMP was defined as the average membrane potential over 4 s with no current input. Spike peak and half-width were determined from a single suprathreshold response to a 20-ms depolarizing current, where peak potential was the maximum voltage and half-width was the width of the spike at a voltage halfway between RMP and the peak. Sag ratio was determined in response to a 40-pA hyperpolarizing current step and defined as the ratio between the peak decrease and steady-state decrease in voltage. Voltage threshold was defined as the highest voltage reached without spiking and was determined from responses to a series of 3-s duration current steps with progressively increasing amplitude. Rheobase was defined as the current required to initiate a spike and was determined using a current ramp increasing linearly from 0 to 150 pA over 2 s. Input resistance was determined as the slope of the line fitted to the voltage increase resulting from current injections between 0.0016 and 0.048 nA.

The modelled cell received simulated synaptic input from conjunctive cells. Synapses were randomly localized to dendritic locations with nine synapses per input cell. The probability of synapse placement on a dendrite was given by the ratio of the dendrite length to the total basal or apical dendritic length. Synapses generated fast conductance changes with an instantaneous rise and exponential decay of 2 ms (cf. ref. [53]). All synapses had the same maximal conductance. We evaluate a 'low-conductance' version of the model in which maximal synaptic conductance was 0.603 nS, and a 'high-conductance' version in which the maximal synaptic conductance was 250 nS.

**Simulation of conjunctive cell input models**. To simulate firing of conjunctive cells, we first generated for each cell a grid pattern and a head direction tuning curve. For the grid pattern, the centre of each field was specified from the vertices of equilateral triangles with a length of 50 cm that were aligned to tessellate the simulated environment. Each cell had the same spatial phase. Each firing field was described by a circular Gaussian distribution with a full width at half maximum (FWHM) of 20.0 cm. To generate non-uniform field maxima (cf. ref. [24]), the peak for each field was scaled by a random value from a uniform distribution between 0 and 1. The grids were then scaled to have a peak height of 1. The head direction tuning curve consisted of a Gaussian centred around a preferred direction with an FWHM of 141 degrees and a height of 1. The preferred head directions were evenly distributed between 0 and 360 degrees. For each simulated conjunctive cell, firing probability distributions were calculated from the grid pattern and a head direction tuning curve at a spatial resolution of 1 cm. The probability of firing was determined as $0.28 \times$ the probability of grid $\times 0.12 \times$ the probability of head direction firing. These values resulted in peak spatial firing rates of around 12 Hz in the low-conductance version and 28 Hz in the high-conductance version. Peak head direction firing rates were around 5 Hz in the low-conductance version and 10 Hz in the high-conductance version, which are comparable to experimental data[8].

Conjunctive cell spike times were generated for a behavioural trajectory within a $1\,\text{m} \times 1\,\text{m}$ open field. The $x$ and $y$ position was determined every millisecond, and if the probability of each cell firing was greater than a random number between 0 and 1, the cell spiked. For 100 simulated conjunctive cells, the peak spatial firing rate was $12.3 \pm 2.71$ Hz, and the normalized peak head direction firing rate was $4.94 \pm 1.04$ Hz.

Each conjunctive cell was connected at nine synapses to the downstream stellate cell. To evaluate representations generated in the stellate neuron model, we simulated 20 trials for each model configuration. Each trial differed in the randomly determined synapse placement of each conjunctive input and in the randomly determined peaks of the conjunctive cell fields.

**Reporting summary**. Further information on research design is available in the Nature Research Reporting Summary linked to this article.

## Data availability
All data will be made available from the Nolan Lab repository on the University of Edinburgh's DataShare site (https://datashare.is.ed.ac.uk/handle/10283/777, https://doi.org/10.7488/ds/2855).

## Code availability
All analysis code will be made available from the Nolan Lab GitHub site (https://github.com/MattNolanLab/grid_cell_analysis).

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

## Acknowledgements

We thank Gülşen Sürmeli and Derek Garden for comments on the manuscript; Emma Wood, Michael Allerhand and members of the Nolan laboratory for helpful discussions and Elizabeth Allison for assistance with set up of open field recordings and initial analysis. We thank scidraw.io for the mouse schematic. This work was supported by grants to M.F.N. from the Wellcome Trust (200855/Z/16/Z) and the BBSRC (BB/L010496/1); to I.P. and M.F.N. from the Centre for Statistics at the University of Edinburgh; by a College of Medicine and Veterinary Medicine PhD Studentship, funded by the Thomas Work Fellowship, to K.G. and by the Wellcome Trust (108890/Z/15/Z) Translational Neuroscience PhD programme to I.H. This work made use of resources provided by the Edinburgh Compute and Data Facility.

## Author contributions

K.G. and M.F.N. conceptualized the study. K.G. performed experiments, developed code and performed analyses. J.P. developed the new computational model. J.P. and I.H. implemented and analysed simulations. K.G., I.P. and M.F.N. contributed to statistical design. H.S. performed histology and imaging. B.V. and K.G. performed experiments for the revisions. M.F.N. obtained funding, supervised the project and wrote the manuscript. All authors contributed to review and editing of the manuscript.

## Competing interests

The authors declare no competing interests.
