## [Peer Review File · Nature Communications]

Reviewers' Comments:

Reviewer #1:

Remarks to the Author:

This manuscript reports findings from grid cells in the entorhinal cortex from both rats and mice. Previous studies have reported two types of grid cells – pure grid cells that do not show any directional tuning in their firing, and conjunctive grid cells, which have a secondary correlate of head directionality. The current study reports that pure grid cells actually do have a secondary component of directionality that has been missed in previous studies. Each node in the grid of a pure grid cell has a different head direction that it fires maximally to. But if one averages over all nodes of the grid cell, then there doesn't appear to be any directionality. The authors also postulate that pure grid cells may be constructed from the inputs of conjunctive grid x direction cells.

These findings are interesting and noteworthy. The issue is an important one and would be of interest to a large community of researchers. The manuscript was generally clear and rather short. Because the findings are relatively simple and easy to explain, it might be possible to condense it and convey these findings as a short communication rather than a full-length paper. While the results are interesting, there are some major issues I have with the data and how it was analyzed.

First, the experiments seem to be more a set of analyses that were applied to a data set of recorded grid cells (possibly for some other experiment), rather than planned from the start to address this issue. If the authors were seeking to test this issue directly, then there are a number of experiments they could have designed to better address the issue. For instance, they could have rotated the salient landmark in the box and determine whether the directional tuning also rotated. They also could have recorded the cells a second day and determine whether the directionality was stable across days rather than just across a single session (i.e., 1st half vs. 2nd half of a session).

Some of the directionality shown for individual grid nodes is not very compelling. For example, in Fig. 3B some of the nodes do not show a high degree of directional modulation (e.g., top row, middle (purple); bottom row, nodes for left (brown) and middle (cyan)). It seems that a better description of their firing would be 'directionally modulated' rather than showing strong directionality.

Two issues not addressed are: 1) did every 'classic' grid cell have nodes that were modulated by head direction? 2) for each 'classic' grid cell, what percentage of the nodes were directionally modulated - 90%, 50%? These analyses should be conducted.

Figure 4B. Of the 6 nodes shown for 1st vs. 2nd half analyses, two of them (green and brown) are not very similar (although the brown one is supposedly significant) and two others (cyan and purple) are not statistically significant. Thus, half of the nodes did not reach significance. Similarly, the legend states that "The median [correlation] (0.29 ± 0.31 for mice and 0.16 ± 0.33 for rats) of both distributions differed from zero.....". Why was a non-parametric test used here? While these values may be significant, both correlation values are relatively low – again, it is not a very compelling case.

Head direction scores were calculated using methods usually not conducted this way. Although there is nothing inherently wrong, it would have been better to use circular statistics and define directional strength using a Rayleigh vector. For example, the Watson U2 test was used to compute a direction score, but this test is not a very conservative test. For example, see the paper by Johnson et al. (Hippocampus, 2005; see Fig. 3). In their study many cells passed a Watson U2 test for directionality, but most of them do not look very directional. In particular, cells that had U2 values < 10 did not appear particularly directional. Many of the U2 values reported here in the present study are well < 10.

Another important issue not addressed in the analyses is how does the strength of directionality across different nodes in pure grid cells compare with the strength of directionality observed in conjunctive grid cells?

Also not addressed: was there any pattern to the directions across the different grid nodes within individual cells? Or did the different preferred directions for each grid node appear random? For example, was there a gradual transition from one direction to another as one traveled across different nodes? I can imagine several types of patterns that would be non-random. The authors should look at this issue.

In sum, the data is not entirely compelling. One might argue, however, that because the directionality is sometimes evident, it might be better to describe the grid nodes as weakly modulated by head direction.

Specific issues:

It would be helpful if there were clear and better terminology used for the different types of grid cells. The authors clearly describe conjunctive cells, but when discussing 'regular', non-conjunctive grid cells, the text is sometimes unclear. I would recommend using the phrase 'pure' grid cells, 'classic' grid cells, or 'traditional' grid cells to distinguish them from the conjunctive ones.

The authors used a 1 deg binning resolution for the firing rate histograms, but a different bin resolution for firing frequency plots (yellow boxes). I doubt that the video tracking was that accurate (1 deg) given the distance between the two different colored polystyrene balls attached to either side of the mouse's head. The distance between the two balls is not mentioned, nor is how they are positioned on the animal's head. Further, if the binning is done as shown by the curves in the yellow box, then the histogram plots should be done at the same resolution.

The polar histogram was smoothed by calculating a rolling sum over a 23 deg window. Why 23°? This is a bit of a strange value to use.

Results section: "..... we found 7.3 ± 3.9 and 4.1 ± 4.5 significant bins / cell for mice and rats respectively (Fig. 1e)." These values are out of how many possible bins?

"..... firing rate histograms of grid cells had multiple significant peaks and troughs". I think the authors mean that individual nodes had different peaks and troughs when looked at over each cell's firing pattern, and not that there were multiple peaks for individual nodes. This should be clarified in the text.

Figure 1 refers to parameters 'rejected bars per cell' and 'significant bars per cell.' These terms could be described better in the text.

The y-axis is labelled incorrectly in Fig. 2b. It should say 'cumulative probability'.

What is the significance of the yellow dashed square in Fig. 3B?

Fig. 3 and other figures. What are the firing rates for the individual plots shown? What is the firing rate scale shown by the gray circles?

It looks like there could be some differences between rats and mice in the cumulative probability plots (it looks like mice curves are often shifted to the right compared to rats, suggesting stronger effect),

which the authors did not test for, although it would be difficult to interpret whether this difference was a species or task difference (foraging vs not foraging).

Also, near the bottom of page 4 the authors show that within-field HD correlations are higher than between-field correlations, but this effect seems to be marginal for rats in the Kolmogorov-Smirnov test (at least compared to mice), which was not mentioned.

The sentence below in the Discussion is not clear.

"This explanation is consistent with models in which grid codes originate in conjunctive cells, but differs in that the location-dependence of grid cell firing emerges from spatial non-uniformities in the conjunctive cell firing fields."

Supplementary Materials

In the example cell figures (pg 46-53), many of the smoothed plots (e.g. 'coverage' plot, 'head direction in detected firing fields' plot) appear rotated 90 deg relative to the path/spike plots. Also, why does it sometimes look like the mice were in a cylinder for part of the recording session; i.e., path plots on p. 46 and 48?

Some of these plots are labeled by noting x number of spikes in y number of seconds. This notation is not typical of how these plots are normally labelled. It is unclear what the maximum firing rate is in these plots.

The font size for many of these plots is too small and unreadable.

The spike waveforms on page 53 are not very useful because of the huge noise artifact.

The authors also mention in the main text that they "recorded from X neurons in the MEC of 15 mice," but Table S2 only lists 14 mice, and only 9 of those animals had electrodes in MEC. Where were the electrodes in the other animals? The authors give the impression that all cells were recorded from the MEC.

Reviewer #2:

Remarks to the Author:

In this manuscript, Gerlei et al. make the case that 'pure' grid cells, in the sense that they lack unimodal coding of the animals' head direction, are, counter to the prevailing view, significantly tuned to multimodal combinations of head directions, with tuning that differs between grid fields. Furthermore, they show that such tuning could be explained by input from conjunctive hd x grid cells to 'pure' grid cells, provided that the conjunctive cell inputs exhibit a nonuniform distribution of firing rates across their spatial firing fields. As the authors note, this possibility has received little attention in the literature and, if true, would have interesting and important implications for how the grid code is generated and how grid cells impact downstream readers.

While I acknowledge that this finding would be important to the field, my primary concern with this work is the need to thoroughly account for potentially confounding variables. The primary confound which the authors try to address is spatial location. They do so by using a shuffling procedure to resample spikes in a way which does not depend on heading. This is the primary control for Figures 1 + 3. While this is a good and obvious starting point, the fact that the shuffled distribution measures are typically an order of magnitude less than the recorded data suggests that there might be a difference in the bias of the actual data that is not accounted for by the shuffled distributions. This

bias might arise due to behavioral biases in running speed, head direction, movement direction, etc. at grid vertices that are not captured by the shuffled control but also contribute to firing rate differences. One could easily imagine a systematic difference in the running distributions based on the location of field (fields at the center would have a more uniform distributions of velocity).

Major comments:

- 1) "Variation in running speed between different parts of the environment is also unlikely to account for directional tuning as firing of most grid cells was independent from running speed (Supplemental Data)". I couldn't find these data in the manuscript and I think this is an important potential confound that needs to be addressed more thoroughly. I also don't immediately agree with the statement that this would be unlikely. For example, we know that the firing rate of grid cells are speed modulated. If running speed systematically changes by head direction with a given field, one would expect that firing rate of that neuron would look directionally modulated in that field even though the directionality can be explained by the animal's behavior.
- 2) Fig 2 + 4 Is chance correlation 0 in this case, because biases in head direction across halves look correlated and may yield a non-zero chance correlation? A shuffled control might help here.
- 3) Is the extent of a bias in head direction within a field correlated the head direction tuning bias in that field, even if the preferred directions differ?
- 4) The authors state "Directional firing remained uncorrelated between fields when considering only fields adjacent to the walls of the arena, or only fields in the centre of the arena, indicating that the location dependence of directional tuning is also not related to the proximity of fields to the borders of the arena." This is a useful analysis, but I am still not convinced. When the authors look at all grid fields across all recordings, is there a systematic relationship between location and directional tuning? i.e. Are cells located close to the north border more tuned to north?
- 5) Is firing rate modulation more specific to head direction than movement direction?
- 6) Page 4: (median correlation for mice: 0.016 ± 0.36 ; for rats: -0.12 ± 0.27) (Fig. 5a-b), \Downarrow median correlation looks positive in the rat figure; am I misreading this?
- 7) The authors also assert that other models of grid cells would not be sufficient to account for the stable multimodal head direction tuning of 'pure' grid cell activity which they observe. While I share their intuition that we might not expect such tuning a priori, I think it is important to show, at least for some standard grid cell models such as the attractor model of Burak and Fiete. I think such simulations could also help address some of my above concerns about confounding variables by testing whether or not biases in the behavioral paths can drive similar effects in models in which pure grid cells are explicitly not modulated by head direction (Burak and Fiete might not be the best model here for that).
- 9) Do model predictions differ according to grid scale/field size, given that the success of the model is dependent on local variations of input conjunctive cell firing rates? Are any of these predictions born out by the data (I realize the mouse data might have a limited distribution of scales, but the rat data should have at least a few modules worth exploring).
- 10) Was head direction tuning by grid field consistent across recording sessions when grid were held across sessions?

Minor comments:

- 1) The authors state they recorded 324 MEC cells in 15 mice, and 38/324 were grid cells. They later state that they analyzed 13 grid cells from 4 mice, and 25 grid cells from rats. This totals 38 grid cells from both mice and rats. I assume this means they recorded 324 cells from 15 mice, only 4 mice produced 13 grid cells while the other 25 grid cells were recorded in rats. Please clarify in the text.
- 2) 4 cells were considered to be conjunctive HD by grid cells. Were these from rats or mice? Were these included in subsequent analyses?
- 3) The authors are missing scale bars throughout the figures including in firing rate maps and circular histograms. Other scale bars are no labeled.
- 4) In figure 1d and 3c are the authors showing SEM or SD?
- 5) Y-axis on Figure 2b should read "Proportion of cells", or "Cumulative probability" to be consistent with other CDFs.
- 6) Supplemental Data 2 includes many unlabeled axes and the text of those labeled is nearly impossible to read.

Reviewer #3:

Remarks to the Author:

In their paper 'Grid cells encode local head direction' Gerlei et al. describe local head direction coding in entorhinal grid cells in rats and mice. The authors find that grid cells firing is not homogeneous across grid cell firing nodes and that different firing nodes may differ in head direction tuning. The relationship of head-direction tuning and grid cell activity has been studied before, but the distinct head direction tuning of individual nodes is a potentially interesting and novel result. There are of course numerous caveats that need to be resolved before the distinct head direction tuning of individual nodes can be firmly established. It seems to me that the authors have considered many of the issues, but they did communicate their results very clearly. My specific concerns are the following:

Caveats

Do different firing nodes may differ in head direction tuning? In general I believe the author's findings and that there is such a thing as differential head direction tuning of individual firing nodes. In particular the stability of the effect across sessions (Figure 4) seems to indicate that the result is not just a fluke. Having said so, I am not sure that the authors ruled out low-level trivial explanations for their findings. A particular concern are differential self-motion effects across between firing notes. If a node is at the wall, one might expect the animal to transverse it with high speeds in directions parallel to the wall, whereas the animal will be slower in directions orthogonal to the wall. Thus, pure speed tuning could give the false impression of a differential head direction tuning between border and mid field firing nodes in such cases. The example is meant to indicate that the authors need to do a more careful analysis of differential self-motion effects.

Raw data and measurements

The assessment of head-direction tuning in individual firing nodes comes with special problems. The major worry is that the data become too thin for a meaningful assessment of head-direction. For this

reason it would be very important to show raw spike plots and trajectory data as it is done in most entorhinal papers. Yet, in this paper, where this information is actually crucial to assess how many spikes and how many traversals the authors have per node, is information is actually missing. Overall it might make sense in this case to perform a number of 'super-long' recording sessions to obtain a very good sampling.

The authors seem very much captured in their own analysis world

The paper in its current form is hard to read. It's not that the authors did not do reasonable work, it is more that they do not seem to expect a non-expert reader. For example Figure 3 evolves around 'bars' per field, whereby the reader can only guess, what the authors mean by 'bar'. The duplication of rat and mouse data is also somewhat distracting. Finally, in Figure 6 the author state 'Direction-dependence of grid cell firing is accounted for by integration of co-aligned, non-uniform conjunctive cell inputs'. I guess they intend to say that their pattern of results could be explained by a certain model, but the flat-out way of putting reflects the broader problem of not thinking outside the own explanation schemes.

Visualization

A better visualization of firing node head direction tuning might be helpful. If the authors could show the strength and (main) direction of tuning of head direction for bunch of cells and firing nodes this might be helpful for the reader to visualize the main effect better. The very detailed maps of few cells shown in the paper do not represent this information easily accessibly.

We thank the reviewers for their helpful comments.

**Reviewer #1 (Remarks to the Author):**

**This manuscript reports findings from grid cells in the entorhinal cortex from both rats**
**and mice. Previous studies have reported two types of grid cells – pure grid cells that do**
**not show any directional tuning in their firing, and conjunctive grid cells, which have a**
**secondary correlate of head directionality. The current study reports that pure grid cells**
**actually do have a secondary component of directionality that has been missed in**
**previous studies. Each node in the grid of a pure grid cell has a different head direction**
**that it fires maximally to. But if one averages over all nodes of the grid cell, then there**
**doesn't appear to be any directionality. The authors also postulate that pure grid cells**
**may be constructed from the inputs of conjunctive grid x direction cells.**

**These findings are interesting and noteworthy. The issue is an important one and would**
**be of interest to a large community of researchers. The manuscript was generally clear**
**and rather short. Because the findings are relatively simple and easy to explain, it might**
**be possible to condense it and convey these findings as a short communication rather**
**than a full-length paper. While the results are interesting, there are some major issues I**
**have with the data and how it was analyzed.**

We appreciate that the reviewer recognises the interest and importance of our results.

We realise now that the previous version of the manuscript was insufficiently clear about the key
findings of the study. In particular, it appears the manuscript could have been read to imply that
individual fields from a grid cell are tuned to a single head direction, while the average over all
nodes does not show directionality. This is an over-simplification. First, our results show that
pure grid cell firing is directionally tuned when all nodes are considered together, but in contrast
to conjunctive cells there is no clear preference for a single head direction (Figures 1 and 2).
Second, directional tuning of individual fields is also typically multidirectional rather than being
tuned to a single direction (Figure 3).

To address this in the revised manuscript we have clarified the importance of distinguishing
between three models for the relationship between neuronal activity and head direction:
omnidirectional firing in which neurons are equally active in all directions (Fig. 1a(i));
unidirectional tuning in which neurons are active in a single preferred direction (Fig 1a(ii)); and
multidirectional tuning in which neurons are preferentially active in several directions (Fig.
1a(iii)). Thus, neurons with firing that differs from the null hypothesis of being independent from
head direction (Fig. 1a(i)), could be tuned to a single direction (Fig. 1a(ii)) as is the case for
conjunctive cells, or could be multidirectional (Fig. 1a(iii)) as we find here for pure grid cells.

*(1) First, the experiments seem to be more a set of analyses that were applied to a data set*
*of recorded grid cells (possibly for some other experiment), rather than planned from the*
*start to address this issue. If the authors were seeking to test this issue directly, then*
*there are a number of experiments they could have designed to better address the*
*issue. For instance, they could have rotated the salient landmark in the box and*
*determine whether the directional tuning also rotated. They also could have recorded the*
*cells a second day and determine whether the directionality was stable across days*
*rather than just across a single session (i.e., 1st half vs. 2nd half of a session).*

Our focus here is on testing the hypothesis that grid cells are directionally tuned by
distinguishing between the three models for the relationship between neuronal activity and head
direction that we outline above.

Our experiments were in fact designed to address this hypothesis, for example by using longer
recording durations and acquiring more spikes than in typical grid cell experiments (recording
duration: our data, 25.1 +/- 3.8 minutes, previous rat data, 13.9 +/- 4.9 minutes, $p < 0.0001$ t-test;
number of spikes: our data, 12,822 +/- 16,375, previous rat data, 2,005, +/- 1,397, $p < 0.0001$
t-test, all data stated as mean +/- SD). It is because of these design features that directional
tuning of pure grid cells is easier to detect in our data compared with the previous rat data. To
make this clearer we now include the recording durations within the main results section (p 2-3,
lines 69-71). We now also include analyses of simulation experiments that clarify the
advantages obtained from the longer duration recordings (Fig. S14).

As the reviewer highlighted earlier, our finding of local directional tuning of pure grid cells is of
interest to a large community of researchers. We appreciate the additional suggestions for
further experiments but it's unclear how these experiments would test the hypothesis that
motivates this finding, that is that firing by pure grid cells is directionally tuned. For example, cue
rotation experiments could be used to address interesting questions about the identity of the
sensory signals that determine the dependence of pure grid cell firing on head direction, but
whether directional tuning does or does not follow rotation of a given cue would not argue for or
against the directional hypothesis we address here. Similarly, recordings over several days
could potentially address interesting questions about the stability of directional tuning, but again,
just as place cell instability does not argue for or against the existence of place cells (cf.
(Kentros et al. 2004)), whether directional tuning turns out to be stable or not over days would
not speak directly to the directional hypothesis we address here. Thus, while these are
interesting suggestions, they will not address the question of whether pure grid cells show
directional firing. Because each suggestion requires testing of mechanistic hypotheses that are
independent from the hypothesis we address here, our opinion is that adding experiments that
address these questions would very likely confuse rather than strengthen the present
manuscript. Therefore, we have instead focused on additional analyses and data that address
the directional hypothesis directly.

(2) *Some of the directionality shown for individual grid nodes is not very compelling. For*
*example, in Fig. 3B some of the nodes do not show a high degree of directional*
*modulation (e.g., top row, middle (purple); bottom row, nodes for left (brown) and middle*
*(cyan). It seems that a better description of their firing would be 'directionally modulated'*
*rather than showing strong directionality.*

Our analyses show that the directional dependence of pure grid cell firing is a substantial effect
that is highly unlikely to have arisen by chance (e.g. $p < 10^{-16}$ for population level analyses of
global and local directional tuning in Figs. 2c and 3d). Our evidence for this conclusion, obtained
initially from experiments with mice, is replicated in an independent rat dataset, and is
accounted for by a well constrained circuit model.

We wonder if the suggestion that “*Some of the directionality shown for individual grid nodes is*
*not very compelling*” arises from an expectation that directional tuning would necessarily be
unidirectional as is the case for classic head direction cells and for conjunctive cells (cf. Fig
1a(ii)). As we outlined above, we have modified the text to be clearer that directional tuning can
also be multidirectional (cf. Fig. 1a(iii)). In addition to making this conceptual issue clearer (e.g.
Fig. 1a), we have replaced the previous classic ‘head direction plots’ with plots that compare the
data with the mean and 10 - 90 % confidence intervals of the corresponding shuffled distribution
(Figures 2, 3 and 5). It is hopefully clear from inspection of these plots that the experimental
data differs substantially from expectations generated by the shuffled data. As a further control
for this analysis, we now include similar plots generated from simulation of previous grid cell
models with experimental trajectories as inputs (Fig. 6). In these plots the simulation results fall
within the shuffled data, demonstrating that these models do not account for the experimental
data, and further validating the analysis approach.

With respect to the field highlighted by the reviewer, the firing rate histogram has 5 bins that
differ significantly ($p < 0.05$) from the shuffled data after correcting for multiple comparisons
(shown here in Reviewer Figure 1 below and in Fig. 3c of the revised manuscript). The number
of significant bins in the experimental data falls completely outside the number generated from
shuffled distributions indicating that it is unlikely to have arisen by chance ($p < 0.001$ based on
1000 shuffles). Thus, although the field does not have a unimodal directional tuning profile it is
nevertheless directionally tuned.

We note that we couldn't find ‘strong directionality’ mentioned in the previous version of the
manuscript, whereas we do consistently refer to modulation by head direction. We have now
modified the manuscript to be clear that directionality refers to any form of directional tuning and
not only to the unimodal tuning that is characteristic of classic head direction cells (e.g Fig. 1a,
lines 44-46).

**Reviewer Figure 1.** Shuffled analysis for a field with multi-modal directional tuning.
 The polar plot (left) of firing binned by head direction has several peaks that differ significantly
 from the shuffled distribution ($p < 0.05$ after correcting for multiple comparison). These peaks
 are marked with an asterisk (*). Comparison of the firing rate in each bin (small filled circles)
 with corresponding shuffled firing rates reveals bins for which the experimental data are outside of
 the 90 % confidence intervals (error bars) of the shuffled distribution (right).

*(3) Two issues not addressed are: 1) did every 'classic' grid cell have nodes that were*
 *modulated by head direction? 2) for each 'classic' grid cell, what percentage of the*
 *nodes were directionally modulated - 90%, 50%? These analyses should be conducted.*

Almost all pure grid cells had nodes with at least one bin from their head direction histogram
 that differed significantly from the corresponding shuffled distribution after correcting for multiple
 comparisons (mice: 12 / 13; rats 24 / 25) (cf. Figure 3c, d).

For each pure grid cell, the proportion of fields with at least one histogram bin that differed
 significantly from the shuffled data after correcting for multiple comparison was 84.6 ± 27.1 %
 for mice (38 / 44 fields in 13 cells) and 61.7 ± 28.2 % for rats (47 / 83 fields in total in 25 cells).

We have added this data to the results (p 5 , lines 136 - 140). We have also added similar
 analyses for the circuit model we introduce that accounts for our results (Fig. 7d). The
 proportions are in good agreement relative to the recording time and number of spikes recorded
 in our experiments.

*(4) Figure 4B. Of the 6 nodes shown for 1st vs. 2nd half analyses, two of them (green and*
 *brown) are not very similar (although the brown one is supposedly significant) and two*
 *others (cyan and purple) are not statistically significant. Thus, half of the nodes did not*

*reach significance. Similarly, the legend states that “The median [correlation] ($0.29 \pm$*
*0.31 for mice and 0.16 ± 0.33 for rats) of both distributions differed from zero.....”. Why*
*was a non-parametric test used here? While these values may be significant, both*
*correlation values are relatively low – again, it is not a very compelling case.*

The data shown in the previous Fig. 4b were chosen to be representative. Just as the proportion
of significantly correlated fields in the population is < 1 (see cumulative distributions now in Fig.
5g), the example showed a mixture of significant and non-significantly modulated fields.

The median correlation referred to by the reviewer includes fields that were not significantly
modulated by head direction. If one considers only the fields with significant direction
modulation then the median correlations are 0.39 ± 0.32 ($n = 35$ fields) for mice and 0.22 ± 0.35
($n = 46$ fields) for rats. When only including significant correlations of directionally modulated
fields, the median values are 0.46 ± 0.32 ($n = 29$ fields) for mice and 0.32 ± 0.38 ($n = 36$ fields)
for rats. The lower correlation scores obtained for rats are consistent with the shorter recording
durations and fewer spikes in the rat data set. Importantly, the proportions that we identify in our
experimental data are in agreement with ‘ground truth’ simulations results in which local
directional tuning is introduced into the models by design (cf. Fig. S14).

We use a non-parametric test here and elsewhere as this makes fewer assumptions about the
underlying data. Making the same comparison using a one sample t-test, we obtained similar
results, again finding that the mean of within-field correlations significantly differed from zero
(mice: $p = 4.10 \times 10^{-7}$, rats: $p = 5.35 \times 10^{-5}$). We keep the non-parametric test in the manuscript
as it is more general.

To address this point in the revised manuscript we have improved the presentation of the
examples. We have also reorganised the ordering of the manuscript so that the consideration of
stability at global and local levels is integrated and follows the initial demonstration of directional
firing of pure grid cells. Because global stability implies local stability and vice versa we hope
this makes the presentation more coherent.

*(5) Head direction scores were calculated using methods usually not conducted this way.*
*Although there is nothing inherently wrong, it would have been better to use circular*
*statistics and define directional strength using a Rayleigh vector. For example, the*
*Watson U2 test was used to compute a direction score, but this test is not a very*
*conservative test. For example, see the paper by Johnson et al. (Hippocampus, 2005;*
*see Fig. 3). In their study many cells passed a Watson U2 test for directionality, but most*
*of them do not look very directional. In particular, cells that had U2 values < 10 did not*
*appear particularly directional. Many of the U2 values reported here in the present study*
*are well < 10 .*

The reviewer's comment here is perhaps confusing directional tuning in general (cf. Fig. 1a(ii
and iii) with unimodal directional tuning (cf. Fig. 1a(ii only)) associated with classic head
direction cells and with conjunctive cell firing. We haven't reported Rayleigh test results as we
were interested in directionality in general, whereas the Rayleigh test aims to distinguish a
unimodal (cf. Fig. 1a(ii)) from a uniform circular distribution (cf. Fig. 1a(i)).

The Watson U^2 test evaluates homogeneity on two samples of circular data. To demonstrate
that the test is appropriately conservative we have applied it to simulated data in which each of
two groups is drawn from the same distribution so as to generate values of the test statistic
expected by chance (Reviewer Figure 2). The values obtained are consistent with the
significance thresholds we use in the manuscript. They are more than 10 fold smaller than the
test statistics obtained from our assessment of direction-dependence of firing by pure grid cells
(cf. Fig. 1c). Thus, it is highly unlikely that these results were obtained by chance. It is possible
that the postsubiculum cells with $U^2 < 10$ reported in Johnson et al. show multidirectional tuning
which would be functionally important, although the analyses did not address the distributive
hypothesis as an alternative explanation (cf. Muller et al. 1994).

**Reviewer Figure 2. Simulated test statistics for the Watson two sample test.**
(a-c) Distributions of the test statistic obtained from comparisons of two groups drawn from the
same uniform circular distribution (a), the same von Mises distribution (b) and the same sum of
three von Mises distributions (c). Each histogram shows results from comparison of 1000
simulated datasets.

*(6) Another important issue not addressed in the analyses is how does the strength of*
*directionality across different nodes in pure grid cells compare with the strength of*
*directionality observed in conjunctive grid cells?*

Because directional tuning of conjunctive cells is unidirectional (cf. Fig. 1a(ii)) whereas
directional tuning of pure grid cells and their firing fields is typically multidirectional (cf. Fig.
1a(iii)) it is difficult to establish a metric that enables direct comparison of the strength of their
directionality. Typically, comparisons of the strength of directionality between two unidirectional
tuning curves would involve comparing the population mean vector of the two distributions.
However, distributions with multiple modes will have a smaller population mean vector even if
there is strong modulation; in an extreme case a distribution with two oppositely oriented modes
would have a population mean of zero. Therefore, this approach is not useful in this case.

To try to address the question we have instead compared the proportion of fields with significant
bins in their distributive plots and the number of significant bins per field (Fig. 3, Fig. S8). The
proportion of fields with significant bins was similar for pure grid cells and conjunctive cells from
mice (grid: 84.6 ± 27.1 %, $n = 44$ fields, 13 cells; conjunctive: $100 \% \pm 0$ %, $n = 3$ fields, 1 cell)
and rats (grid: 61.7 ± 28.3 %, $n = 83$ fields, 25 cells; conjunctive: 65.9 ± 31.8 %, $n = 24$ fields, 7
cells). For individual fields from pure grid cells, the average number of significant bins was 4.27
± 3.15 for the observed data vs 0.15 ± 0.12 for the shuffled data for mice ($n = 44$ fields) and 2.07
± 2.83 vs $0.012 / 20 \pm 0.11$ for rats ($n = 83$ fields). For conjunctive cells, the average number of
significant bins was 16.33 ± 0.47 in the observed data vs 0.012 ± 0.13 in the shuffled data for
mice ($n = 3$ fields) and 4.29 ± 4.09 in observed data in rats ($n = 24$ fields) vs 0.012 ± 0.12 in
shuffled data. These analyses suggest that the proportion of directional fields is similar between
pure and conjunctive grid cells, but conjunctive cells have a greater number of directions in their
firing field that differ significantly from their corresponding shuffled distribution. The latter
observation may simply be a result of unimodal tuning of conjunctive cells versus multimodal
tuning of pure grid cells.

We have added these data to Fig S8 and refer to them on p 5, line 50 - 55.

*(7) Also not addressed: was there was any pattern to the directions across the different grid*
*nodes within individual cells? Or did the different preferred directions for each grid node*
*appear random? For example, was there a gradual transition from one direction to*
*another as one traveled across different nodes? I can imagine several types of patterns*
*that would be non-random. The authors should look at this issue.*

We previously looked for organisation by comparing centrally located fields with fields around
the border of the arena. We have now tested for patterns in two further ways. First, for pairs of
fields from the same cell, we examined the relationship between their separation and the
Pearson coefficient obtained from correlation of their head direction histograms (Fig. S10a).
This analysis did not reveal evidence for a statistically significant relationship. Second, for each

pair of fields from the same cell, we rotated one of the head direction histograms in one degree
steps and correlated each rotated histogram with the other head direction histogram. We then
identified the angle with the highest correlation (Fig. S10b). The angle corresponding to the
highest correlation did not show any statistically significant dependence on the distance
between the fields.

Fig. S10 showing results of these analyses is referred to on p. 6 lines 174-176 of the main text.

*(8) In sum, the data is not entirely compelling. One might argue, however, that because the*
*directionality is sometimes evident, it might be better to describe the grid nodes as*
*weakly modulated by head direction.*

We believe that the evidence for directional modulation of pure grid cell firing is compelling. The
evidence that the firing of pure grid cells from mice and rats are globally tuned to head direction
is extremely unlikely to be explained by the null hypothesis of no directional tuning (e.g. for
population level analysis in Fig. 2c, $p < 10^{-16}$), as is the evidence that individual firing fields are
directionally tuned (e.g. for population level analyses in Fig. 3d, $p < 10^{-16}$ vs the null hypothesis
of no directional tuning).

We recognise from the comments made that the previous version of the manuscript was
insufficiently clear about the ways in which directional tuning may manifest, which could have
made it hard to appreciate the extent of directional modulation. We have outlined in our
responses above changes made to the manuscript that aim to make the key results clearer. We
hope that this revised presentation is more compelling.

**Specific issues:**

*(9) It would be helpful if there were clear and better terminology used for the different types*
*of grid cells. The authors clearly describe conjunctive cells, but when discussing*
*'regular', non-conjunctive grid cells, the text is sometimes unclear. I would recommend*
*using the phrase 'pure' grid cells, 'classic' grid cells, or 'traditional' grid cells to*
*distinguish them from the conjunctive ones.*

We now use the term 'pure grid cell' to distinguish classic grid cells from conjunctive cells.

*(10) The authors used a 1 deg binning resolution for the firing rate histograms, but a*
*different bin resolution for firing frequency plots (yellow boxes). I doubt that the video*
*tracking was that accurate (1 deg) given the distance between the two different colored*
*polystyrene balls attached to either side of the mouse's head. The distance between the*

*two balls is not mentioned, nor is how they are positioned on the animal's head. Further,*
*if the binning is done as shown by the curves in the yellow box, then the histogram plots*
*should be done at the same resolution.*

The 1 deg bin used for the classic head direction histograms (Fig. 1b, Fig. 5b,f) is of course
over-sampled, which is why the data was subsequently smoothed. We based these analyses on
scripts provided by colleagues as they are a standard plot used by other groups.

The results of the analyses that use shuffled data are shown with a 18 degree bin size, giving
20 bins in total. This is the size of the binning used for calculations of the distribution of the
shuffled data. The bin size was chosen to maximise the power of the analyses, by having many
data points per bin, while maintaining sensitivity to different head directions.

We have updated the Methods to clarify the points above and to provide details of the extraction
of the reference points from the video image (p 27, lines 347 - 365).

*(11) The polar histogram was smoothed by calculating a rolling sum over a 23 deg*
*window. Why 23°? This is a bit of a strange value to use.*

We received this analysis function from one of our collaborators and other than the window size
being a little unusual we did not have any reason to modify it. We show below examples of polar
plots using different window sizes (Reviewer Figure 3). The window size does not affect the
conclusions one would draw from the data.

**Reviewer Figure 3. Examples of polar plots of head direction with different smoothing**
**window sizes.** Binned counts of head direction across all video frames (black) and the mean
firing rate for each head direction bin (red) plotted as a function of head direction. The data were
smoothed using 10, 20, 30 and 40 degree windows (plots from left to right).

*(12) Results section: “..... we found 7.3 ± 3.9 and 4.1 ± 4.5 significant bins / cell for*
*mice and rats respectively (Fig. 1e).” These values are out of how many possible bins?*

There are 20 bins in the shuffled analyses, each covering 18 degrees. We have modified the
text to make this clear.

(13) *“..... firing rate histograms of grid cells had multiple significant peaks and troughs”.*
*I think the authors mean that individual nodes had different peaks and troughs when*
*looked at over each cell’s firing pattern, and not that there were multiple peaks for*
*individual nodes. This should be clarified in the text.*

This sentence refers to the polar head direction histogram that includes spikes from all fields of
the grid cell (Figure 1). We do mean that this field histogram had multiple peaks (see further
discussion above).

(14) *Figure 1 refers to parameters ‘rejected bars per cell’ and ‘significant bars per cell.’*
*These terms could be described better in the text.*

In the revised manuscript we no longer use the ‘rejected bars per cell’ metric as we realise now
that it is potentially confusing and is effectively redundant. We have included an additional
schematic, modified the figure legend and changed the text to clarify the description of
‘significant bars per cell’.

(15) *The y-axis is labelled incorrectly in Fig. 2b. It should say ‘cumulative probability’.*

We have updated the axis label.

(16) *What is the significance of the yellow dashed square in Fig. 3B?*

The yellow box marks the head direction polar plot of the firing field that corresponds to the
histogram shown as an example of the shuffled analysis in 3c. We added more text to the
legend to make this clearer.

(17) *Fig. 3 and other figures. What are the firing rates for the individual plots shown?*
*What is the firing rate scale shown by the grey circles?*

We have added the maximum firing rates above all polar plots in the manuscript. The grey
circles were not firing rates, but were the head directions from the trajectory. Where appropriate
we have now replaced these with the mean and 10 - 90 % range of the shuffled data as this is
the relevant null distribution to which the experimental data should be compared.

(18) *It looks like there could be some differences between rats and mice in the cumulative*
*probability plots (it looks like mice curves are often shifted to the right compared to rats,*
*suggesting stronger effect), which the authors did not test for, although it would be*
*difficult to interpret whether this difference was a species or task difference (foraging vs*
*not foraging).*

The analyses we present were designed to test whether there is directional firing in pure grid
cells from either mice or rats but not to compare the two species. New experiments would be
required to make this comparison. For example, such experiments would need to control for
exploration time, type of enclosure, type of behaviour (foraging or no foraging) and use the
same spike sorting algorithm. A possible reason for the bigger effect we see in the mouse data
is that the average time spent in fields (mice: 80.89 ± 64.23 s, $n = 76$ fields; rats: 29.58 ± 22.65 ,
$n = 83$ fields; data are mean \pm SD) and the number of spikes per field (mice: 881.53 ± 698.33 , n
$= 76$ fields; rats: 275.98 ± 275.53 , $n = 83$ fields) is significantly higher in the mouse data (for
time spent in fields $p = 7.68 \times 10^{-5}$; for number of spikes per field $p = 8.58 \times 10^{-14}$, 2 sample
Kolmogorov-Smirnov test). Our simulations show that this increase in the amount of data used
for the analysis will make the directional tuning appear larger (Figure S14).

(19) *Also, near the bottom of page 4 the authors show that within-field HD correlations*
*are higher than between-field correlations, but this effect seems to be marginal for rats in*
*the Kolmogorov-Smirnov test (at least compared to mice), which was not mentioned.*

The effect in rats is significant ($p = 0.018$, $D = 0.53$) but is small. This is expected given the
much smaller number of spikes in the rat data (see comments above and Figure S14). We now
briefly highlight this in the main text (p 6, line 104 - 105).

(20) *The sentence below in the Discussion is not clear. "This explanation is consistent*
*with models in which grid codes originate in conjunctive cells, but differs in that the*
*location-dependence of grid cell firing emerges from spatial non-uniformities in the*
*conjunctive cell firing fields."*

We mean that in existing models where grid cells inherit their firing patterns from conjunctive
cells, conjunctive cells have similar mean firing rates across all of their fields, while in our model
the mean firing rate differs between fields. We have modified the text in this section to try to
improve the clarity.

*Supplementary Materials*

(21) *In the example cell figures (pg 46-53), many of the smoothed plots (e.g. 'coverage'*
*plot, 'head direction in detected firing fields' plot) appear rotated 90 deg relative to the*

*path/spike plots. Also, why does it sometimes look like the mice were in a cylinder for*
*part of the recording session; i.e., path plots on p. 46 and 48?*

We have rotated the figures to correct the plotting error.

For the path plots on p 46 and p 48 the mice were running in circles in those specific cases for
part of the session. All mouse recordings were done in the same rectangular box that is
described in the methods.

(22) *Some of these plots are labeled by noting x number of spikes in y number of*
*seconds. This notation is not typical of how these plots are normally labelled. It is*
*unclear what the maximum firing rate is in these plots.*

We added these numbers to indicate the sampling of individual firing fields. We have now also
added the maximum firing rates to all appropriate plots.

(23) *The font size for many of these plots is too small and unreadable.*

We have increased the font size for the supplementary figures.

(24) *The spike waveforms on page 53 are not very useful because of the huge noise*
*artifact.*

We have modified the supplementary figures to make waveforms more visible.

(25) *The authors also mention in the main text that they "recorded from X neurons in the*
*MEC of 15 mice," but Table S2 only lists 14 mice, and only 9 of those animals had*
*electrodes in MEC. Where were the electrodes in the other animals? The authors give*
*the impression that all cells were recorded from the MEC.*

The data reported in the manuscript is from 8 mice. For 7 of these mice the tetrode location was
confirmed as within the MEC and for the other mouse the tetrode location could not be
determined. To obtain the data we implanted a total of 16 mice targeting the MEC. Two animals
had to be terminated before the end of the experiment and six animals did not have grid firing
fields and so did not contribute to the reported dataset. We include these numbers to make
clear the proportion of animals in which we were able to identify grid cells. We have clarified
these numbers in the Results and Methods section of the revised manuscript.

**Reviewer #2 (Remarks to the Author):**

**In this manuscript, Gerlei et al. make the case that ‘pure’ grid cells, in the sense that they**
**lack unimodal coding of the animals’ head direction, are, counter to the prevailing view,**
**significantly tuned to multimodal combinations of head directions, with tuning that**
**differs between grid fields. Furthermore, they show that such tuning could be explained**
**by input from conjunctive hd x grid cells to ‘pure’ grid cells, provided that the**
**conjunctive cell inputs exhibit a nonuniform distribution of firing rates across their**
**spatial firing fields. As the authors note, this possibility has received little attention in the**
**literature and, if true, would have interesting and important implications for how the grid**
**code is generated and how grid cells impact downstream readers.**

**While I acknowledge that this finding would be important to the field, my primary concern**
**with this work is the need to thoroughly account for potentially confounding variables.**
**The primary confound which the authors try to address is spatial location. They do so by**
**using a shuffling procedure to resample spikes in a way which does not depend on**
**heading. This is the primary control for Figures 1 + 3. While this is a good and obvious**
**starting point, the fact that the shuffled distribution measures are typically an order of**
**magnitude less than the recorded data suggests that there might be a difference in the**
**bias of the actual data that is not accounted for by the shuffled distributions. This bias**
**might arise due to behavioral biases in running speed, head direction, movement**
**direction, etc. at grid vertices that are not captured by the shuffled control but also**
**contribute to firing rate differences. One could easily imagine a systematic difference in**
**the running distributions based on the location of field (fields at the center would have a**
**more uniform distributions of velocity).**

We appreciate the recognition of the novelty and important implications of the results.

The reviewer suggests three hypotheses that might account for the difference between the
experimental data and the shuffled data, referred to by the reviewer as a bias. These are: head
direction, which is the primary hypothesis that we consider here; movement direction, which is
usually closely related to but not necessarily the same as head direction; and running speed.
We will briefly summarise new analyses that further address confounding variables including
movement direction and running speed. We will provide details in our responses to the specific
points below.

The alternative hypothesis that variation in running speed accounts for local directional
modulation is important to consider and did not receive sufficient attention in the previous
version of the manuscript. We now show that for the majority of pure grid cells in our dataset
running speed modulation of their firing is weak or absent. This is consistent with previous
publications (Kropff et al. 2015). Importantly, directional modulation of pure grid cell firing does
not correlate with running speed modulation and is clearly present in neurons with speed scores

that are effectively zero. Thus, systematic differences in running speed can not explain local
directional modulation of grid firing.

We now also show that firing of pure grid cells also correlates with movement direction but the
effect is weaker than for head direction. The presence of a correlation for movement direction is
expected as movement direction and head direction are closely correlated. Because the
relationship does not appear as strong for movement direction, head direction is likely to be the
main contributor to the directional modulation of pure grid cell firing.

To further assess the robustness of our analyses we have included new simulations of various
previous grid models. Importantly, these simulations use experimentally recorded behavioural
trajectories as inputs. These simulations do not generate location-dependent directional firing,
therefore demonstrating that behavioural biases in combination with standard grid models do
not account for the experimental data.

**Major comments:**

(26) 1) *“Variation in running speed between different parts of the environment is also*
*unlikely to account for directional tuning as firing of most grid cells was independent from*
*running speed (Supplemental Data)”. I couldn’t find these data in the manuscript and I*
*think this is an important potential confound that needs to be addressed more*
*thoroughly. I also don’t immediately agree with the statement that this would be unlikely.*
*For example, we know that the firing rate of grid cells are speed modulated. If running*
*speed systematically changes by head direction with a given field, one would expect that*
*firing rate of that neuron would look directionally modulated in that field even though the*
*directionality can be explained by the animal’s behavior.*

Modulation of the firing of pure grid cells in our dataset by running speed is weak in most cells
(median speed score for mouse pure grid cells = 0.068 ± 0.18 , $n = 34$, range = $-0.060 - 0.48$;
median for rat pure grid cells from the Moser lab dataset = 0.038 ± 0.048 , $n = 68$) and speed
modulated neurons are relatively rare (mouse 8 / 34 neurons with speed score > 0.1 ; rat 8 / 68
neurons with speed score > 0.1). This is consistent with previous reports (e.g. Kropff et al.
2015). This suggests that changes in running speed are unlikely to explain
directional-dependence of pure grid cell firing fields. We show the dependence of firing rate on
running speed for all neurons that were used for analysis of firing fields in Reviewer Figure 4
below (see also panels for each neuron in the Supplemental Data).

To address the issue further, we have directly examined the relationship between running speed
and directional modulation of individual fields. The proportion of directionally modulated fields
and the number of significant bins per field either do not significantly differ or are in fact higher
for neurons with low speed scores (< 0.1) compared to those with high speed scores (\geq
0.1)(mice: 9 / 13 fields with speed scores ≥ 0.1 are directionally modulated with 2.7 ± 1.4

significant bins / field, whereas 29 / 31 fields with speed scores < 0.1 are directionally
modulated with 5.6 ± 2.9 significant bins / field, $p = 4.9 \times 10^{-4}$ for Mann-Whitney U test on number
of directional bins; rats: 7 / 13 fields with speed scores ≥ 0.1 are directionally modulated with 3.4
± 3.2 significant bins / field, whereas 49 / 70 fields with speed scores < 0.1 are directionally
modulated with 3.7 ± 2.9 significant bins / field, $p = 0.364$ for Mann-Whitney U test on number of
directional bins). This is the opposite to what we would expect if modulation of firing rate by
running speed causes local directional tuning. Furthermore, when we consider only neurons
with speed scores $< \pm 0.05$, we again find the directional modulation of their firing fields remains
robust (mice: 9 / 10 fields with speed scores $< \pm 0.05$ are directionally modulated with 6.2 ± 3.1
significant bins / field; rats: 29 / 53 fields with speed scores < 0.05 are directionally modulated
with 3.4 ± 2.9 significant bins / field).

We note that speed modulated neurons are nevertheless detectable in our dataset but they
typically do not have grid-like spatial firing fields and their head direction plots show little sign of
directional modulation (Reviewer Figure 5).

Together, these analyses show that speed modulation is unable to explain the directional
modulation of firing in our dataset. To address this in the manuscript, we now include summary
statistics for speed modulation of the pure grid cells in the main text (p 4, line 80 - 84 and Fig.
S5) and present speed plots for all grid cells (Supplementary Data 2).

**Reviewer Figure 4. Speed modulation is weak in most recorded grid cells.** Firing rate as a
function of running speed for each grid cell recorded from mice that included in the analysis of
individual firing fields (Fig 3). Speed scores, calculated as the r value from linear regression of
the firing rate as a function of speed (Kropff et al. 2015) are shown above the plots.

**Reviewer Figure 5. Examples of speed modulated cells recorded from the mouse**
 **entorhinal cortex.** Spike waveforms, firing rate as a function of running speed, open field firing
 rate heat map, and directional firing (red) compared to time in each direction (black) for two
 speed cells. Values of the Watson U^2 test statistic (0.74 and 0.78) are below the threshold for
 statistical significance (see Reviewer Figure 3 above and Fig. 1d).

*(27) 2) Fig 2 + 4 Is chance correlation 0 in this case, because biases in head direction*
 *across halves look correlated and may yield a non-zero chance correlation? A shuffled*
 *control might help here.*

Given the conclusion that the distributive hypothesis (cf. Muller et al., 1994) does not explain the
 directional tuning of pure grid cell fields, which is supported by our analyses in the revised Fig. 2
 and 3 and our new analyses of previous grid models (Fig. 6), then we expect the chance
 correlation to be zero. Nevertheless, to address this further we now include additional controls
 using shuffled data following the reviewer's suggestion (revised Fig. 5). We also include
 validation of these analyses using simulated data (Fig S14). These analyses continue to support
 the conclusion that directional tuning is stable within a recording session.

*(28) 3) Is the extent of a bias in head direction within a field correlated the head direction*
 *tuning bias in that field, even if the preferred directions differ?*

This does not appear to be the case. We have quantified the within field bias in the trajectory
 using the test statistic from the one sample Watson test and the head direction tuning bias using
 the number of significant bins for data binned as a function of head direction. We did not

observe a relationship between the two (mice: slope = 0.035, $p > 0.1$, rats: slope = -0.00061, $p >$
0.1)(Fig. S7). We now show these analyses in Fig S7 And refer to them in the Results section (p
5, line 109-110).

(29) 4) *The authors state “Directional firing remained uncorrelated between fields when*
*considering only fields adjacent to the walls of the arena, or only fields in the centre of*
*the arena, indicating that the location dependence of directional tuning is also not related*
*to the proximity of fields to the borders of the arena.” This is a useful analysis, but I am*
*still not convinced. When the authors look at all grid fields across all recordings, is there*
*a systematic relationship between location and directional tuning? i.e. Are cells located*
*close to the north border more tuned to north?*

We show below the distributive plots for individual fields relative to their position within the
environment (Reviewer Figure 6).

With respect to the directional hypothesis that we test here, the key observation is that fields in
the central sector are directionally modulated and do not show any consistent preferred set of
directions. This argues against alternative hypotheses based on directional modulation being
related to proximity to the wall.

The additional question raised by the reviewer is whether there is any systematic organisation of
directional tuning relative to the boundaries of the arena. In sectors adjacent to the walls
different fields also show different directional modulation. There is no obvious pattern to the
directional organisation. We now show these data as Fig. S9 And refer to them from p 6, line
44-46.

**Reviewer Figure 6. Relationship between directional tuning and location in the arena.**

Distributive plots for all isolated mouse grid fields organised according to the location of the
 centre of each field in the recording arena.

 *(30) 5) Is firing rate modulation more specific to head direction than movement direction?*

 To determine whether firing rate modulation is more specific to head direction than movement
 direction, we calculated movement direction (Raudies et al. 2015) and repeated the shuffled
 analyses presented in the manuscript for head direction. We found that fewer grid cells were
 significantly modulated by movement direction relative to head direction (26 / 34 grid cells

modulated by movement direction and 34 / 34 modulated by head direction for mice, and 41 / 68
modulated by movement direction grid cells and 56 / 68 head direction modulated grid cells for
rats). The number of significantly modulated bins in distributive histograms was lower relative to
the head direction analysis (movement direction: 2.5 ± 3.5 bins / cell for mice and 2.6 ± 3.9 bins
657 / cell for rats; head direction: in mice: 7.3 ± 3.9 bins / cell, rats: 4.1 ± 4.5 bins / cell). When we
analyzed individual fields we found that slightly more fields were modulated by movement
direction than head direction in mice, but fewer in rats (43 / 44 grid fields in 13 grid cells
modulated by movement direction and 38 / 44 by head direction in mice, and 39 / 83 fields in 25
cells modulated by movement direction and 47 / 83 by head direction in rats). The number of
significantly modulated bins in distributive histograms was similar to the head direction analysis
results in mice and there were fewer significant bins in rats (movement direction: 4.5 ± 2.9 bins /
field for mice and: 0.98 ± 1.3 bins / field for rats vs head direction in mice: 4.3 ± 3.2 bins / field;
rats: 2.1 ± 2.8 bins / field).

Together, these results suggest that the effects of movement direction on directional firing were
somewhat smaller than head direction. We now refer to these analyses in the Results (p 4, lines
108 - 110) and show them in Fig. S3.

(31) 6) Page 4: (median correlation for mice: 0.016 ± 0.36 ; for rats: -0.12 ± 0.27) (Fig.
5a-b), median correlation looks positive in the rat figure; am I misreading this?

This was a typo. The median for rats is 0.12 and not -0.12.

(32) 7) The authors also assert that other models of grid cells would not be sufficient to
account for the stable multimodal head direction tuning of 'pure' grid cell activity which
they observe. While I share their intuition that we might not expect such tuning a priori, I
think it is important to show, at least for some standard grid cell models such as the
attractor model of Burak and Fiete. I think such simulations could also help address
some of my above concerns about confounding variables by testing whether or not
biases in the behavioral paths can drive similar effects in models in which pure grid cells
are explicitly not modulated by head direction (Burak and Fiete might not be the best
model here for that).

We greatly appreciate this suggestion. We also agree that simulations from these models can
further mitigate the concerns raised about confounding variables. We have now simulated two
oscillatory interference models (Giocomo et al. 2007 and Burgess et al. 2007) and two
continuous attractor models (Guanella et al. 2007 and Pastoll et al. 2013). Importantly, we have
used experimentally recorded behavioural trajectories as inputs to the models.

None of the simulated previous models are able to account for the experimental data. The
closest is the Pastoll et al. model, which generates a similar proportion of directionally
modulated fields to the experimental data, but with far fewer significant bins per field.

As the reviewer suggested, these simulations provide a further important control for potential
confounding variables. All of the simulations used experimentally recorded trajectories as inputs
and inspection of their directional firing rate plots shows that their firing clearly falls within the
range predicted by the corresponding shuffled data. This provides further evidence against
interpretations of the experimental data that rely on confounding behavioural variables. We have
also extended our analysis of the model with spatially non-uniform conjunctive cell inputs as a
positive control to provide further validation for our analyses.

To address this point in the revised manuscript we now show examples of simulated grid fields
and their directional tuning for the Burgess et al. 2007 and the Pastoll et al. 2013 models as a
main figure (Fig. 6) and for the Giocomo et al. 2007 and Guanella et al. 2007 models as a
supplemental figure (Fig. S11). We also show comparison of summary statistics for the models
with the data (Fig 6) and extended analysis of the conjunctive cell input model (Fig S14).

(33) 9) Do model predictions differ according to grid scale/field size, given that the
success of the model is dependent on local variations of input conjunctive cell firing
rates? Are any of these predictions born out by the data (I realize the mouse data might
have a limited distribution of scales, but the rat data should have at least a few modules
worth exploring).

We have carried out additional simulations with the non-uniform conjunctive cell input models.
These simulations don't identify any effect of field spacing ($r = 0.079$, $p = 0.34$) and show a
weak but statistically significant effects of field size ($r = -0.12$, $p = 0.0091$). Although this effect is
detectable with a large number of simulations, there is insufficient experimental data to detect
an effect of this size. These results are now shown in Fig. S15.

(34) 10) Was head direction tuning by grid field consistent across recording sessions
when grid were held across sessions?

In the experiments reported in the manuscript, electrodes were lowered after every recording to
minimise the chance of double-reporting data from the same cell, so we did not collect data that
would allow us to test this. In response to suggestions from Reviewer #3 we made additional
recordings with 60 minutes duration sessions (see below). These recordings show that global
and local directional firing is robust across these longer recording durations. However, they also
show that at this time scale grid fields appear to shift their locations. This spontaneous shift of
the fields was unexpected as grid fields are not usually investigated with recording sessions of
this length. The shift in field locations precludes analysis of location-dependence of their

directional firing between early and late parts of the long session. Similar concerns preclude
analyses across sessions.

*Minor comments:*

(35) *1) The authors state they recorded 324 MEC cells in 15 mice, and 38/324 were grid*
*cells. They later state that they analyzed 13 grid cells from 4 mice, and 25 grid cells from*
*rats. This totals 38 grid cells from both mice and rats. I assume this means they*
*recorded 324 cells from 15 mice, only 4 mice produced 13 grid cells while the other 25*
*grid cells were recorded in rats. Please clarify in the text.*

We apologise for the confusion. We recorded from 16 mice, but only identified grid cells in 8
mice. The analysis of global directional tuning uses 39 grid cells from mice (5 conjunctive and
34 pure grid cells) and 100 grid cells from rats (32 conjunctive and 68 pure grid cells). The
analysis of local directional firing was limited to cells for which we could automatically detect at
least two firing fields. This was the case for 13 pure grid cells from mice and 25 pure grid cells
from rats.

We updated the text at relevant parts of the manuscript to clarify this. The reason for reporting
all animals is to be clear about the proportion of mice where we successfully identified as having
detectable grid cells.

(36) *2) 4 cells were considered to be conjunctive HD by grid cells. Were these from rats*
*or mice? Were these included in subsequent analyses?*

The four conjunctive cells referred to here are from mice and were used in the analysis where
classic head direction plots of pure and conjunctive grid fields are compared to the other fields
within the same cell to assess how similar they are to each other. The conjunctive cells were
only included in these analyses (Fig. 4).

(37) *3) The authors are missing scale bars throughout the figures including in firing rate*
*maps and circular histograms. Other scale bars are no labeled.*

We have updated the scale bars and indicated the maximum firing rates on the relevant figures.

(38) *4) In figure 1d and 3c are the authors showing SEM or SD?*

This is the 90 % confidence interval. It's stated in the figure legend.

(39) 5) Y-axis on Figure 2b should read “Proportion of cells”, or “Cumulative probability” to
be consistent with other CDFs.

We have corrected this.

(40) (6) Supplemental Data 2 includes many unlabeled axes and the text of those labeled
is nearly impossible to read.

We have modified the presentation of these figures so they are clearer to read.

**Reviewer #3 (Remarks to the Author):**

**In their paper ‘Grid cells encode local head direction’ Gerlei et al. describe local head**
**direction coding in entorhinal grid cells in rats and mice. The authors find that grid cells**
**firing is not homogeneous across grid cell firing nodes and that different firing nodes**
**may differ in head direction tuning. The relationship of head-direction tuning and grid**
**cell activity has been studied before, but the distinct head direction tuning of individual**
**nodes is a potentially interesting and novel result. There are of course numerous caveats**
**that need to be resolved before the distinct head direction tuning of individual nodes can**
**be firmly established. It seems to me that the authors have considered many of the**
**issues, but they did communicate their results very clearly. My specific concerns are the**
**following:**

**Caveats**
**Do different firing nodes may differ in head direction tuning? In general I believe the**
**author’s findings and that there is such a thing as differential head direction tuning of**
**individual firing nodes. In particular the stability of the effect across sessions (Figure 4)**
**seems to indicate that the result is not just a fluke. Having said so, I am not sure that the**
**authors ruled out low-level trivial explanations for their findings. A particular concern are**
**differential self-motion effects across between firing notes. If a node is at the wall, one**
**might expect the animal to transverse it with high speeds in directions parallel to the**
**wall, whereas the animal will be slower in directions orthogonal to the wall. Thus, pure**
**speed tuning could give the false impression of a differential head direction tuning**
**between border and mid field firing nodes in such cases. The example is meant to**
**indicate that the authors need to do a more careful analysis of differential self-motion**
**effects.**

Our initial analysis demonstrated that directional tuning at the level of individual fields is unlikely
to be a chance result (e.g. for population level analyses in Fig. 3d, $p < 10^{-16}$ vs the null

hypothesis of no directional tuning) and showed that speed tuning was rare in pure grid cells
(Supplemental Data). To further address the possibility that speed tuning causes the directional
tuning we now include additional analyses (see response 26 above). These analyses
demonstrate that directional tuning does not correlate with speed tuning and is present in the
absence of detectable modulation of firing by running speed. In addition, we now include
simulations of previous grid cell models using experimentally recorded trajectories as inputs
(see response 32 above). These simulations are unable to account for local directional tuning
providing further evidence against modulation of firing by behavioural parameters causing
directional tuning.

**Raw data and measurements**

*(41) The assessment of head-direction tuning in individual firing nodes comes with*
*special problems. The major worry is that the data become too thin for a meaningful*
*assessment of head-direction. For this reason it would be very important to show raw*
*spike plots and trajectory data as it is done in most entorhinal papers. Yet, in this paper,*
*where this information is actually crucial to assess how many spikes and how many*
*traversals the authors have per node, this information is actually missing. Overall it might*
*make sense in this case to perform a number of 'super-long' recording sessions to*
*obtain a very good sampling.*

We have shown raw spike plots and trajectories in the Supplemental Data for all included pure
grid cells we recorded from mice. We have modified these figures to make them clearer and
easier to interpret. We also show an example trajectory in Fig. 7a.

We have also carried out additional 60 minute recordings (compared to an average of 25
minutes for our previous recordings and 12 minutes for the rat recordings). We show below an
example pure grid cell from one of these experiments. The global firing of this neuron is strongly
directionally modulated (Reviewer Figure 7) and all identified firing fields were significantly
directional (in the 100th percentile relative to shuffled data))(Reviewer Figure 8a). Thus,
directional modulation of firing is maintained over long recording sessions. This experiment also
reveals unexpected challenges in analysis of data from long recording sessions. Most strikingly,
the firing fields show considerable rate remapping over time and the locations within the fields
with the highest firing rate shift (Reviewer Figure 8b). While this phenomenon is interesting, it
precludes further analysis of directional firing as we don't yet have a statistical framework for
accounting for these time-dependent changes in the location of grid cell activity.

**Reviewer Figure 7. Pure grid cell properties from a one hour long recording.** Top row left
to right: action potential waveforms overlaid for the four channels of the tetrode,
autocorrelograms of spike times, histogram of firing times over time, histogram of speed plotted
against firing rate. Second row left to right: trajectory of the animal (black line) and firing events
(red dots), firing rate map, autocorrelation matrix for rate map, classic polar head direction
histogram (cell firing; red; movement: black). Third row left to right: Detected firing fields on the
rate map and classic polar head direction histograms for the detected fields.

a

b

**Reviewer Figure 8. Local directional modulation and spatial instability in a one hour long**
**recording.** (a) Distributive head direction histograms for the fields in Reviewer Figure 6,
comparing observed (colours) and shuffled (grey) firing rates as a function of head direction.
Stars indicate significantly directional head directions. (b) Firing rate maps from the first and
second halves of the session and the two rate maps subtracted from each other. The firing rate
changes in the subtracted plot reflect shifting of the location of the grid fields during the long
recording session.

(42) *The authors seem very much captured in their own analysis world. The paper in its*
*current form is hard to read. It's not that the authors did not do reasonable work, it is*
*more that they do not seem to expect a non-expert reader. For example Figure 3*
*evolves around 'bars' per field, whereby the reader can only guess, what the authors*
*mean by 'bar'. The duplication of rat and mouse data is also somewhat distracting.*
*Finally, in Figure 6 the author state' Direction-dependence of grid cell firing is accounted*
*for by integration of co-aligned, non-uniform conjunctive cell inputs'. I guess they intend*
*to say that their pattern of results could be explained by a certain model, but the flat-out*
*way of putting reflects the broader problem of not thinking outside the own explanation*
*schemes.*

We have worked hard to improve the readability of the manuscript. Changes made include a
new schematic to illustrate the hypotheses addressed by our analyses (Fig. 1a), a new
schematic to illustrate the motivation for the shuffling analysis (Fig. 2a), and a new
representation of directional firing in which data, the control bin, and directions that differ from
the control distribution are shown (Figs. 2, 3, 4, 6, 7, S2, S6, S9). We have also included
additional explanations throughout the main text.

The duplication of rat and mouse data provides important support for our conclusions. Our initial
conclusions were based on analysis of the data from mice. Validation of our results with data
from rats shows that the results can not be explained by overly flexible initial analyses, and
demonstrates that our conclusions generalise to a second species.

We have also modified the text to try to improve the description of the new model.

(43) *Visualization*

*A better visualization of firing node head direction tuning might be helpful. If the authors*
*could show the strength and (main) direction of tuning of head direction for bunch of*
*cells and firing nodes this might be helpful for the reader to visualize the main effect*
*better. The very detailed maps of few cells shown in the paper do not represent this*
*information easily accessibly.*

We have replaced the previous cartesian plots with polar plots in which head direction tuning is
compared to the control shuffled distribution and in which significant differences are indicated
(Figures 2, 3, 4, 6, 7, S2, S6, S9). We have also improved the explanation to be clear that
directional tuning can be both unidirectional, as is the case for head direction and conjunctive
cells, and multi-directional as we find here for most pure grid cells.

We show directional tuning plots for all grid cells in the Supplemental Data as well as examples
of multiple individual nodes from one cell in Fig. 3. The population level extent of directional
tuning, which is very large, is summarised in Fig. 2c and Fig. 3d.

References

Kentros, Clifford G., Naveen T. Agnihotri, Samantha Streater, Robert D. Hawkins, and Eric R. Kandel. 2004. "Increased Attention to Spatial Context Increases Both Place Field Stability and Spatial Memory." *Neuron* 42 (2): 283–95.

Kropff, Emilio, James E. Carmichael, May-Britt Moser, and Edvard I. Moser. 2015. "Speed Cells in the Medial Entorhinal Cortex." *Nature* 523 (7561): 419–24.

Raudies, Florian, Mark P. Brandon, G. William Chapman, and Michael E. Hasselmo. 2015. "Head Direction Is Coded More Strongly than Movement Direction in a Population of Entorhinal Neurons." *Brain Research* 1621 (September): 355–67.

Reviewers' Comments:

Reviewer #2:

Remarks to the Author:

The authors have completed substantial additional analysis and modeling that sufficiently address the concerns I outlined in the first round of review.

Reviewer #3:

Remarks to the Author:

The authors responded very thoroughly to my (and the other referee's) questions. The revisions of the authors improved the paper. I am more convinced now that there is differential directionality of grid nodes. The revision also consolidated my impression, however, that such directionality differences are quite small. The small effect size is a concern as it touches upon the core message. With all that I am weakly supportive of publication. In case the paper is published I'd argue to weaken the presentation of conclusions and not to oversell the grid node directionality.

Reviewer #2 (Remarks to the Author):

The authors have completed substantial additional analysis and modeling that sufficiently address the concerns I outlined in the first round of review.

We thank the reviewer for the feedback.

Reviewer #3 (Remarks to the Author):

The authors responded very thoroughly to my (and the other referee's) questions. The revisions of the authors improved the paper. I am more convinced now that there is differential directionality of grid nodes. The revision also consolidated my impression, however, that such directionality differences are quite small. The small effect size is a concern as it touches upon the core message. With all that I am weakly supportive of publication. In case the paper is published I'd argue to weaken the presentation of conclusions and not to oversell the grid node directionality.

We thank the reviewer for the feedback. To address the reviewer's suggestion we have revised the title, abstract and main text to state that grid fields are 'locally modulated by head direction', as opposed to 'encode local head direction' or are 'tuned to head direction'. This is also consistent with suggestions made by Reviewer 1 in their comments on the initial manuscript.